# Improving Reliability in Reconstruction of Landsat EVI Seasonal Trajectory over Cloud-Prone, Fragmented, and Mosaic Agricultural Landscapes

Wei Xue [1,2,*], Jonghan Ko [3], Ruyin Cao [4] and Zhiguo Yu [5]

1   Key Laboratory of Oasis Ecology of Education Ministry, College of Ecology and Environment, Xinjiang University, Urumqi 830049, China
2   Xinjiang Jinghe Observation and Research Station of Temperate Desert Ecosystem, Ministry of Education, Jinghe 833300, China
3   Department of Applied Plant Science, Chonnam National University, Gwangju 61186, Republic of Korea; jonghan.ko@jnu.ac.kr
4   School of Resources and Environment, University of Electronic Science and Technology of China, 2006 Xiyuan Avenue, Chengdu 611731, China; cao.ruyin@uestc.edu.cn
5   School of Hydrology and Water Resources, Nanjing University of Information Science and Technology, Nanjing 210044, China; zhiguo.yu@nuist.edu.cn
*   Correspondence: xuewei@lzu.edu.cn; Tel.: +86-0931-8912560

**Abstract:** Although the Landsat 30 m Enhanced Vegetation Index (EVI) products are important input variables in land surface models, recurring Landsat 5/7 EVI time series over cloud-prone, fragmented, and mosaic agricultural landscapes is still a great challenge. In this study, we put forward a simple, but effective "Light and Temperature-Driven Growth model and Double Logistic function fusion algorithm" (LTDG_DL). The empirical basis of the LTDG_DL algorithm was traced from the de Wit crop growth simulation model and the commonly observed nonlinear correlation between the EVI and the Leaf Area Index (LAI). It assimilates the ground daily solar radiation and air temperature to generate seasonal profiles of the empirical LAI and EVI and conducts the within-season calibration of the empirical EVI by adjusting crop growth using cloud-free Landsat EVI observations. The initial date of seedling emergence ($DOY_{ini}$) and the accumulated Growing Degree Days for completion of the vegetative and Flowering stage (FGDDs) were variables to which the algorithm's accuracy was most sensitive. The variable-constrained optimization of the LTDG_DL algorithm was performed by loading the seedling emergence calendar of local prevailing crops and establishing an FGDD lookup table with an exhaustive sampling without replication method. Compared to temporal interpolation functions and Landsat–MODIS spatiotemporal fusion algorithms, the LTDG_DL algorithm had superior performance in the predictions of the EVI increment slope at the vegetative growth stage, the timing of the peak EVI, and the protection of key Landsat EVI observations over cloud-contaminated and complex landscape agricultural systems. Finally, the advantages and limitations of the LTDG_DL algorithm are discussed.

**Keywords:** Landsat EVI; time series; EVI reconstruction; agricultural landscapes

## 1. Introduction

In the 2000s, land holdings per family in China were recorded at 0.53 ha [1]. In Korea, approximately 69% of farms have small land parcels measuring less than 1.0 ha [2]. The majority of agricultural landscapes in East Asia consist of small land parcels (<1.5 ha, approximately 120 × 120 m) [3–5], which are significantly smaller than the average holding size of 19.3 ha in the USA [6]. Additionally, most patchy and fragmented agricultural landscapes in East Asia practice multi-crop planting [7,8]. One common atmospheric characteristic in East Asia monsoon areas is the regular occurrence of precipitation during the summer season, when crop growth accelerates [9,10]. However, the use of open-access

high-temporal-resolution products from the Moderate-Resolution Imaging Spectroradiometer (MODIS) in highly fragmented and mosaic agricultural landscapes has been limited due to their coarse spatial resolution [7]. On the other hand, satellite products provided by the Landsat series, which are free to the public, offer an improved spatial resolution of 30 m and the longest continuous record of land surface reflectance since 1984 This includes Landsat 5 (1984–2013), Landsat 7 (1999–present), and Landsat 8 (2013–present) [11]. These data make it possible to analyze historical changes in land use and land cover change in the agricultural sector and detect crop types in certain areas [6,12–14]. However, research on the quality of Landsat data has shown that obtaining at least three high-quality observations from Landsat 5–7 within a single crop growing season is not feasible on a large scale due to issues such as cloud coverage, selective scene acquisition, sensor malfunction, and other factors [15–17]. Therefore, monitoring crop growth in cloud-prone and complex landscape regions is still a significant challenge, especially when using retrospective Landsat VI data from Landsat 5/7.

The contemporary methods for reconstructing Landsat VI can be classified into two groups based on the reconstruction algorithms: those that use auxiliary information to infer the reflectance or VI of ground objects and those that do not [18,19]. The first group includes VI/reflectance temporal interpolation functions, while the second group consists of Landsat–MODIS VI/reflectance spatiotemporal fusion methods. While some reconstruction algorithms provide calibrated reflectance estimates, most of the algorithms in the literature focus on VI as the outcome [20]. The VI temporal interpolation function is specifically designed for Landsat VI data. Two commonly used global temporal interpolation functions, namely Savitzky–Golay (SG) and Double-Logistic (DL), have been successfully applied to single-year continuous MODIS VI time series [21–24]. They have also been used for aggregating multi-year cloud-free Landsat Enhanced Vegetation Index (EVI) data [25]. Harmonic time series models, which utilize sines and cosines, have shown good performance in fitting single-year temporally rich Landsat Normalized Difference Vegetation Index (NDVI) data, with the Root-Mean-Squared Error (RMSE) ranging from 0.05 to 0.08 [26]. However, reliable NDVI reconstruction is challenging when fitting single-year coarse Landsat observations, resulting in higher RMSE values (RMSE $\geq$ 0.12) [27]. The coefficient estimations of the global temporal interpolation functions are highly sensitive to the range of dates and the density of Landsat VI observations. These functions also lack flexibility [28]. Although long-term mean changes in Landsat VI can be detected, they cannot adequately represent inter-annual variability in the land cover and land use in the agricultural sector.

In Landsat–MODIS VI spatiotemporal fusion methods, the first step is to perform gap-filling of missing Landsat values in order to obtain more cloud-free satellite data, then one or two smoothing functions are applied [5,19,29–32]. The Landsat–MODIS fusion method predicts Landsat VI data at daily or consistent time steps. It is trained using a medium number of cloud-free Landsat and MODIS NDVI image pairs on base dates, and then, it predicts Landsat images on prediction dates along with corresponding cloud-free MODIS NDVI image. Various Landsat–MODIS fusion methods with different algorithm structures have been developed. Examples include the Gap-Filling and Savitzky–Golay filtering method (GF-SG) [32], the Spatial and Temporal Adaptive Reflectance Fusion Model (STARFM) [33] and its derivatives [19,34], and the Flexible Spatiotemporal DAta Fusion (FSDAF) [35]. The GF-SG/STARFM/FSDAF method involves a key step, which is the generation of a synthesized NDVI time series. This is achieved by fusing MODIS NDVI time series data at 250/500 m resolution with cloud-free Landsat observations (referred to as Landsat observations unless otherwise indicated). In order to improve the fusion accuracy, algorithms have progressed from directly merging multiple satellite photos on distinct forecast dates to the weighted fusion of partially cloud-free images [32,36,37]. However, there are four critical issues that these fusion methods encounter, which need to be addressed: (1) The EVI, which can reduce the saturating effects of dense canopy as much as possible, performs better than the NDVI [38]. (2) The spatial resolution of MODIS pixels

(500/250 m) covers fragmented land parcels with multi-crop planting systems, resulting in week spatial autocorrelation between the MODIS and Landsat VI temporal shapes. This leads to incorrect quantification of the land surface crop phenology under certain conditions. (3) The reliability of these methods depends on a medium number of consistently gap-filled high-quality Landsat data on dates close to the prediction dates, which cannot be guaranteed in cloud-prone areas wherein the whole Landsat scene is contaminated by clouds at several acquisition dates. (4) These methods cannot be applied to regions or years where/when MODIS observations are not available. Consequently, there may be inadequate reconstruction in Landsat EVI time series in certain areas. For example, this can occur with fast-growing crops that have continuous missing MODIS/Landsat EVI values or in fragmented agricultural ecosystems, where crops with diverse phenological features grow adjacent to each other.

Although numerous methods for Landsat EVI reconstruction have been proposed, the question of how to reduce uncertainty in the daily series reconstruction for small land parcels during periods without MODIS data such as the 1990s and 2000s remains unanswered. This study addressed this question by introducing a novel algorithm, Light- and Temperature-Driven Growth and Double-Logistic fusion (LTDG_DL), which uses daily ground meteorological data and Landsat EVI observations to generate daily 30 m EVI profiles for the crop growing seasons. Section 2 provides a detailed description of the conceptual framework of LTDG_DL and its bivariable co-strained optimization solutions (Section 2). Section 3 discusses the validation objects, including four farming parcels in the Republic of Korea, China, and the USA where the crop types, Landsat, and field EVI observations were known and available; the cloud-prone Haean basin in the Republic of Korea, which features fragmented and mosaic agricultural landscapes with a known crop phenology; and the cloud-prone Shandong Province in China, where the crop types in double-cropped agricultural systems are unknown. The Sections 4 and 5 present the results and discussion, respectively.

## 2. Development of the LTDG_DL Algorithm

The LTDG_DL algorithm is executed on a daily basis for reconstructing one-season EVI. It consists of two interconnected sub-models (Figure 1): the DL function and the LTDG model. While LTDG_DL does not require site calibration at the per-pixel level, it is still necessary to refine exclusive parameters for different agroclimatic zones. Two structural parameters must be constrained in order to obtain satisfactory daily Landsat EVI data. The algorithm principles of LTDG_DL and its structural parameters are explained in Section 2.1, followed by Section 2.2, which presents the bivariable-constrained optimization solutions.

### 2.1. Algorithm Principles
2.1.1. The DL Function

The SG and DL functions are commonly used statistical measures for preparing satellite VI time series. The SG function is primarily utilized to smooth consecutive VI time series by applying a weighted moving average with a polynomial of a certain degree [39]. The degree of the smoothing polynomial is determined by the time window size and the number of data points within that window. However, the SG function has limitations when dealing with irregularly distributed observations and continuously missing data points over an extended period. This is due to the fact that rare satellite observations can be considered as "sudden/spike points" and are either removed or flattened [28]. Previous studies have proposed the use of the DL function to fit MODIS NDVI observations from a single season, which allows for less homogeneity in the temporal distribution while preserving important cloud-free observations [23,24]. In this study, we applied the DL model to fit Landsat EVI observations for one season using Equation (1), resulting in the generation of daily EVI data (dEVI$_{DL}$) (as shown in Figure 1).

$$f(t, x_1, x_2, x_3, x_4) = \frac{1}{1 + \exp\left(\frac{x_1 - t}{x_2}\right)} - \frac{1}{1 + \exp\left(\frac{x_3 - t}{x_4}\right)} \tag{1}$$

where $x_1$ and $x_3$ are the locations of the left (the increasing phase) and right (the decreasing phase) inflection points, respectively; $x_2$ and $x_4$ determine the rate of change at these points; $t$ is the day of the year. $x_1$ and $x_3$ are defined as the date when the crop growth rate is maximized and the date when the crop senescence rate is maximized, respectively. The ranges of $x_2$ and $x_4$ are limited to [8.8, 40.9] and [8.8, 40.9], respectively. The DL model, which incorporates constrained values for $x_2$ and $x_4$, has shown the ability to accurately predict VI time series data for temperate crops [24]. It should be noted that the range of $x_2$ and $x_4$ may differ between temperate and tropical crops.

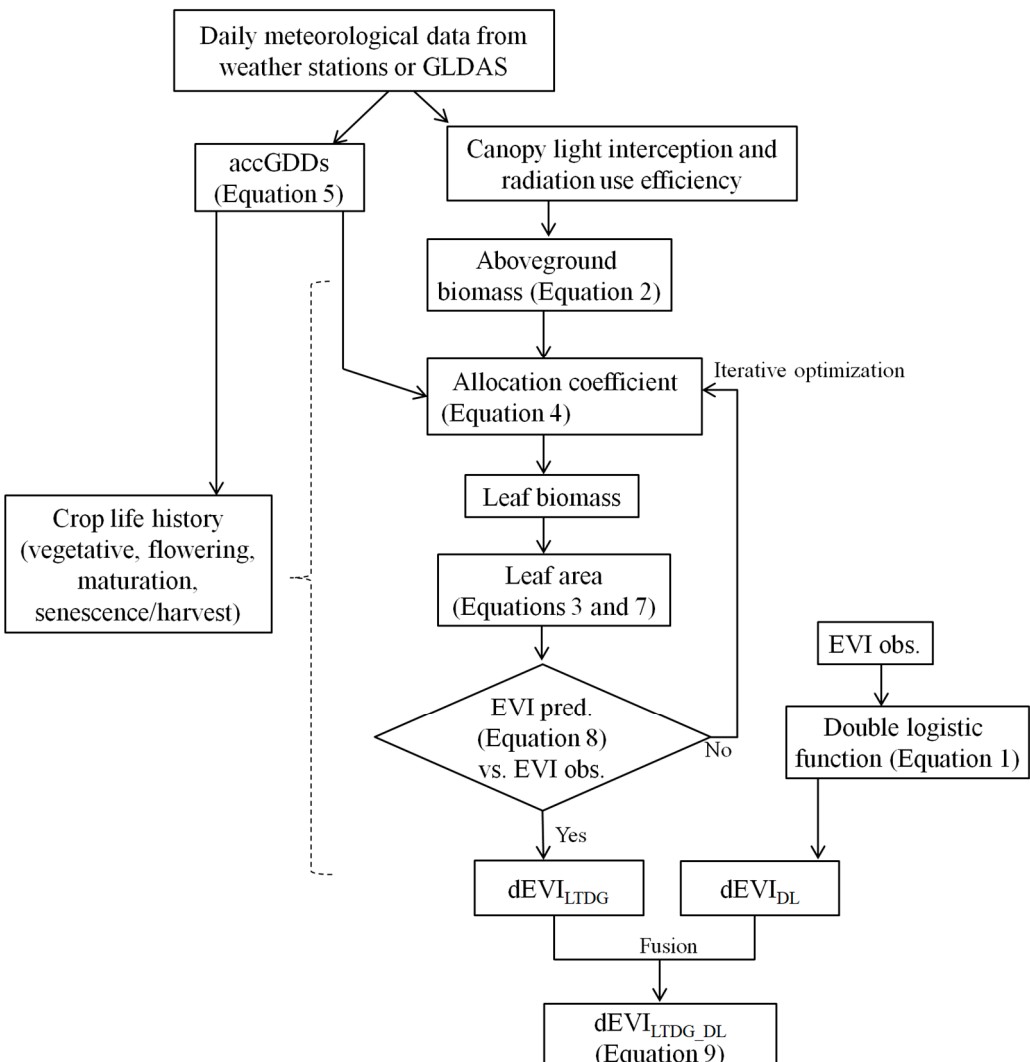

**Figure 1.** Schematic illustration of the LTDG_DL algorithm to reconstruct one-season daily Landsat EVI data in complex landscape agricultural systems. accGDDs: accumulated Growing Degree Days; EVI: Enhanced Vegetation Index; LTDG_DL: Light- and Temperature-Driven Growth (LTDG) model and Double-Logistic (DL) function fusion algorithm; dEVI_LTDG: daily EVI data reconstructed by the LTDG; dEVI_DL: daily EVI data reconstructed by the DL; pred.: prediction; obs.: observation; GLDAS: the Global Land Data Assimilation System; canopy light interception: the amount of solar radiation intercepted by plant canopies.

### 2.1.2. The LTDG Model

The LTDG model explores the principles of the "School of de Wit" crop growth simulation models [40,41]. It operates under the assumptions that the crop growth and development potentials are primarily influenced by the interception of sunlight by the canopy (i.e., the amount of solar radiation intercepted by the canopy) and the accumulation of Growing Degree Days (accGDDs) [42]. The variations observed in satellite EVI measurements throughout the seasons are attributed to changes in crop growth dynamics, specifically alterations in canopy size in irrigated and rainfed fields [38,43–45]. In the LTDG model, the daily aboveground biomass production is estimated using a radiation use efficiency model, which is then multiplied by canopy light interception and radiation use efficiency [46]. Canopy light interception is typically determined using Beer's law, which calculates it as the product of the green Leaf Area Index (LAI) and light extinction coefficient ($k$) [47]. The increase in aboveground biomass on the $i$th day after seedling emergence or transplanting ($\Delta M_{AGB,i}$, units: g) is represented by the following equation:

$$\Delta M_{AGB,i} = \varepsilon \times \beta \times R_{s,i} \downarrow \times (1.0 - \exp(-k \times LAI_i)) \tag{2}$$

where $\varepsilon$ is the radiation use efficiency (default 3.5 g MJ$^{-1}$); $\beta$ is the fraction of total solar radiation that is photosynthetically active radiation (default 0.45); $R_{s,i}\downarrow$ is the daily incident solar irradiance on the $i$th day (MJ day$^{-1}$); $k$ is light extinction coefficient (default 0.5); $LAI_i$ is the green Leaf Area Index on the $i$th day (m$^2$ m$^{-2}$).

The expansion of the canopy area during the vegetative growth and flowering stages, as well as its decline during the maturation stage are regulated by a biomass allocation coefficient ($\rho$) (Equation (3)). The $\rho$ value indicates whether there is positive or negative growth and senescence in the canopy. This coefficient, expressed as a function of accGDDs in Equation (4), determines the amount of daily increase in aboveground biomass allocated to leaves.

$$\Delta LA_i = \rho_i \times \Delta M_{AGB,i} \times SLA \tag{3}$$

$$\rho_i = 1.0 - a_i \times \exp(b_i \times accGDDs_i) \tag{4}$$

where $\rho_i$ is the allocation coefficient of $\Delta M_{AGB,i}$ to leaves on the $i$th day; $SLA$ is the Specific Leaf Area (default 0.024 m$^2$ g$^{-1}$); $\Delta LA_i$ is the daily increase in canopy Leaf Area on the $i$th day (m$^2$ m$^{-2}$); $accGDDs_i$ is the accGDDs on the $i$th day; $a_i$ and $b_i$ are model coefficients on the $i$th day. The GDD on the $i$th + 1 d ($GDD_{i+1}$, °C) is computed by the daily air temperature and base temperature as follows:

$$accGDDs_{i+1} = accGDDs_i + GDD_{i+1} \tag{5}$$

$$GDD_{i+1} = \max[T_{air,i+1} - T_{base}, 0] \tag{6}$$

where $T_{air,i+1}$ is the daily air temperature on the $i$th + 1 d (°C) and $T_{base}$ is the base temperature below which crop growth ceases (default 5 °C for crops in temperate regions). The accGDDs on the initial date of seedling emergence/transplanting (DOY$_{ini}$) are equal to GDD$_1$. The accumulated leaf area on the $i$th + 1 d is a product of the LAI on the $i$th day plus the daily increase in leaf area on the $i$th day (Equation (7)), and the empirical EVI on the $i$th + 1 d is generated by using a generic EVI-LAI equation (Equation (8)).

$$LAI_{i+1} = \Delta LA_i + LAI_i \tag{7}$$

$$EVI_{i+1} = e \times LAI_{i+1}^f \tag{8}$$

where $e$ and $f$ are model coefficients (defaults 0.39 and 0.51, respectively) [41].

The empirical EVI can deviate significantly from the actual Landsat EVI observation on the same date due to the seasonal impacts of environmental factors, such as prolonged drought or nutrient deficiency, affecting crop growth. The LTDG method utilizes Landsat EVI observations to calibrate the empirical EVI within a season. To ensure that the adjusted EVI simulations match the Landsat EVI observations on all observation dates, the daily $\rho$ and initial conditions of the LAI ($LAI_{ini}$, default 0.002 m$^2$ m$^{-2}$) at seedling emergence/the transplanting date are dynamically adjusted using an iterative numerical procedure. In this iterative process, two statistics, $E^+$ and $E^-$, are used to quantify the difference between the simulation and the observation values. $E^+$ represents the sum of the positive errors between the simulation and the observation values; likewise, $E^-$ represents the absolute value of the sum of the negative errors. The total error, E, is calculated as the sum of $E^+$ and $E^-$. A satisfactory fit between the simulation and observation EVI values occurs when $E^+$ is equal to $E^-$. To determine the $LAI_{ini}$ that achieves the best fit, the secant method developed by Conte and de Boor [48] is employed. Once the fit involving $LAI_{ini}$ is obtained, daily $\rho$ is adjusted using the parameters $a_i$ and $b_i$ through parabolic interpolation [49] to minimize the value of E.

The life history of annual and perennial crops is traditionally divided into four phenological phases: vegetative, flowering, grain/fruit maturation, and senescence/harvest phases. The development of the crop phenology closely follows the accumulation of GDDs throughout its lifespan. The accGDDs required to transition from one phenological stage to the next are relatively consistent within crops, but vary greatly among different crop types [42,50]. In this study, the total number of accGDDs needed to complete each phenological phases is represented by VGDDs, FGDDs, MGDDs, and HGDDs, respectively. In determinate crops, the plant stops growing beyond the point where the flowers have bloomed. Conversely, morphologically indeterminate crops such as cotton exhibit simultaneous growth and flower setting, which differs from the determinate crops, such as paddy rice. Considering the use of the LTDG in multi-crop planting systems that include both indeterminate and determinate crops, the VGDDs and FGDDs for the vegetative and flowering phases are combined (hereinafter referred to as FGDDs). Differences in crop phenological development between early-maturing and late-maturating varieties mainly stem from variations in the FGDDs [50]. The FGDDs can be considered the primary variable that differs considerably among crop types.

For agricultural pixels with infrequent Landsat EVI observations over prolonged periods of time, the daily EVI data reconstructed by LTDG ($dEVI_{LTDG}$) may yield higher or lower values than the $dEVI_{DL}$ on certain observation dates. Therefore, an average of the $dEVI_{DL}$ and $dEVI_{LTDG}$ without weighted coefficients is used as the resulting output ($dEVI_{LTDG\_DL}$). This calculation is performed as follows:

$$dEVI_{i,LTDG\_DL} = \sum_{i=1} (dEVI_{i,LTDG} + dEVI_{i,DL})/2.0 \tag{9}$$

### 2.2. Bivariable-Constrained Optimization Solutions

Although LTDG_DL includes numerous parameters, the reconstruction of daily EVI data can be achieved satisfactorily by utilizing constrained configurations of two specific parameters: $DOY_{ini}$ and the FGDDs.

### 2.2.1. Constrained Configurations of $DOY_{ini}$

$DOY_{ini}$ can vary greatly across different agroclimatic zones due to variations in personal preference for seeding date and choice of crop [24]. In agricultural landscapes within agroclimatic zones where the crop types are unknown, a landscape-level $DOY_{ini}$ setting was establishing by incorporating the seedling emergence calendar of locally predominant crops. To configure the landscape-level $DOY_{ini}$, satellite EVI observations from a specific number of agricultural pixels were merged to generate an endmember ensemble of Landsat EVI observation time series (see Figure S1 in the Supporting Information, Section S1). The

onset of the consistent increase phase in the seasonal EVI curves included in the endmember EVI ensemble can be used to determine the landscape-level $DOY_{ini}$ more easily.

### 2.2.2. Constrained Configurations of the FGDDs

Uncertainty in the FGDDs may significantly impact Landsat EVI reconstruction accuracy [41]. In a case study of the paddy rice crop, there were six Landsat EVI observations from $DOY_{ini}$ until the end of the growing seasons (Figure 2a). Reconstructed daily EVI data that well-captured the Landsat EVI observations were obtained using FGDDs = 900 °C. However, when only four Landsat EVI observations were available over the rice growing seasons (two mid-season EVI observations in Figure 2a were masked in Figure 2c), a better reconstruction of the daily EVI profile was achieved using FGDDs = 750 °C. The FGDDs at 750 °C in the case with only three available Landsat EVI observations (three data points in Figure 2a were masked in Figure 2e) generated a superior daily EVI curve compared to other FGDDs (Figure 2e). These results indicate that the satisfactory generation of daily Landsat EVI data using FGDDs is dependent on the number of available Landsat EVI observations. Given the sensitivity of the LDTG_DL modeling performance to the density of Landsat EVI observations, incorporating more EVI data points into LTDG_DL may help improve EVI reconstruction accuracy.

Landsat EVI observations from $DOY_{ini}$ to the end of the crop growing seasons (around mid-October in Northeast Asia) were utilized to generate various combinations of EVI observations. An exhaustive sampling without replication method was employed to develop these groups of EVI observation combinations, resulting in *m* groups of EVI observation combinations, as shown in Figure 3. Each of these *m* groups was then input into LTDG_DL to produce an equal number of $dEVI_{LTDG\_DL}$ outputs. By averaging the *m* groups of $dEVI_{LTDG\_DL}$, a single seasonal EVI profile was obtained (see $\overline{dEVI_{LTDG\_DL}}$ in Figure 3). Notably, it was observed that EVI combination groups with a least three Landsat EVI observations performed better than those with only one or two observations. If there were fewer than three Landsat EVI observations available over the crop growing seasons, no EVI combination groups could be generated. Figure 2b illustrates the generation of twenty EVI combination groups using the exhaustive sampling method with six Landsat EVI observations ($C_6^3 = 20$). Interestingly, the results showed no significant differences in reconstructed daily EVI data when comparing the FGDDs of 750, 800, 850, and 900 °C, which contrasts with the finding presented in Figure 2a. Furthermore, Figure 2d exhibits the superior performance of the FGDDs = 750 °C compared to Figure 2c when four groups of EVI combinations ($C_4^3 = 4$) were used to drive the LTDG_DL model. This is evidenced by the reconstructed maximum EVI closely aligning with the maximum EVI observation shown in Figure 2a. In Figure 2b,d,e, it is apparent that the FGDDs of 750 °C yielded the best results in reconstructing daily EVI data. These findings suggest that the exhaustive sampling without replication method improved the accuracy of EVI reconstruction in crop pixels with low Landsat EVI observation density.

A set of FGDDs produces the same number of $\overline{dEVI_{LTDG\_DL}}$, which were used to create a reference database called $LEVI_{reference,n}$ (refer to Figure 3). The appropriate FGDDs for each agricultural pixel can be determined by comparing them with Landsat EVI observations on various observation dates and selecting the one in the $LEVI_{reference,n}$ database that shows the best agreement with the Landsat EVI observations (i.e., the smallest RMSE), considered as the optimized $dEVI_{LTDG\_DL}$ (see Figure 3). Additionally, a lookup table was established, which comprises *n* groups of FGDDs, MGDDs, and HGDDs (Figure 3). The FGDD lookup table, accompanied by an exhaustive sampling method without replication, plays a vital role in generating the optimized $dEVI_{LTDG\_DL}$.

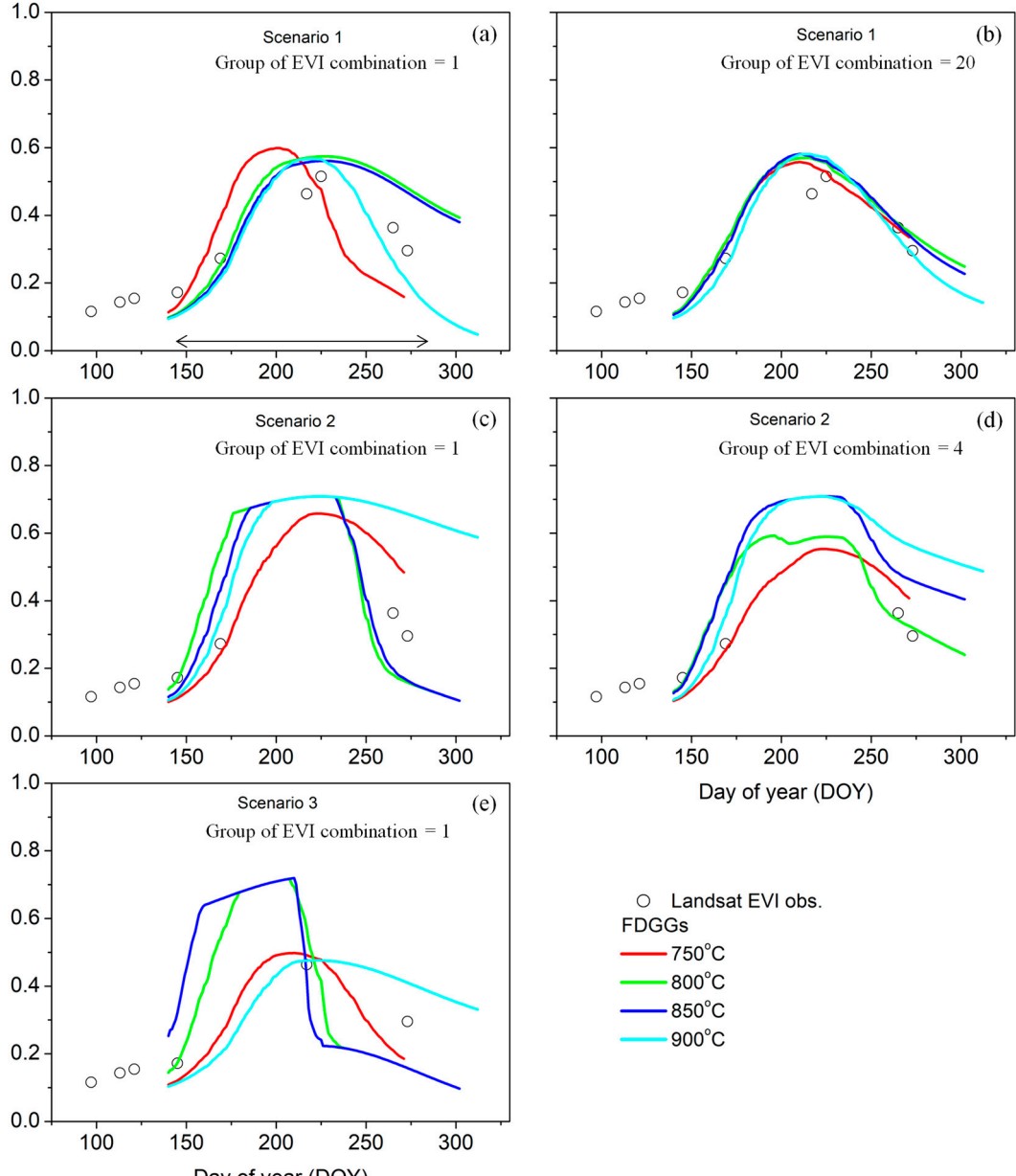

**Figure 2.** Effects of varying FGGDs values (750, 800, 850, and 900 °C, (**a**,**c**,**e**)) and the group number of Landsat EVI observation combinations (**b**,**d**) on Landsat EVI reconstruction. Crop growing seasons are indicated by the double-arrow line in (**a**). Six, four, and three Landsat EVI observations from DOY$_{ini}$ to the end of the crop growing seasons are available in Scenarios 1, 2, and 3, respectively. The group of EVI observation combination = 1 in (**a**,**c**,**e**) represents that all EVI observations of the crop growing seasons were grouped into one EVI observation combination. For the case of six EVI observations over the crop growing seasons, the exhaustive sampling method without replication helped yield 20 groups of EVI observation combinations, wherein each group had at least three EVI observations ($C_6^3 = 20$, (**b**)). Using this procedure, four EVI observations over the crop growing seasons yielded four groups of EVI observation combinations ($C_4^3 = 4$, (**d**)), and three EVI observations over crop growing seasons had only one group of EVI observation combination ($C_3^3 = 1$, (**e**)). FGDDs: the accumulated Growing Degree Days for completion of the vegetative and Flowering stage; EVI: Enhanced Vegetative Index; DOY$_{ini}$: the initial seedling emergence/transplanting date; LTDG_DL: Light- and Temperature-Driven Growth model and Double-Logistic function fusion algorithm; obs.: observation.

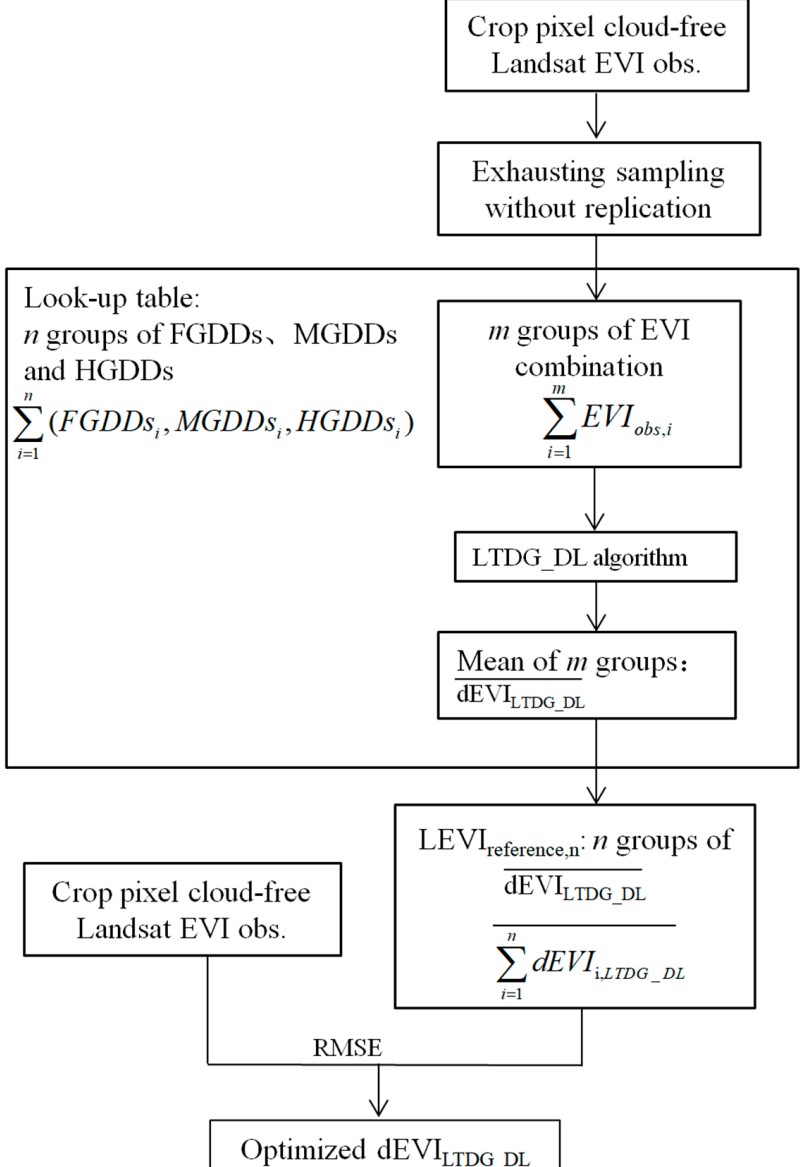

**Figure 3.** Data-processing roadmap of the FGDD-constrained optimization solution involved in the LTDG_DL algorithm. obs.: observation; GDDs: Growing Degree Days; EVI: Enhanced Vegetation Index; RMSE: root mean square error; LTDG_DL: a parameter-constrained Light- and Temperature-Driven Growth (LTDG) model and Double-Logistic (DL) function fusion algorithm; FGDDs: the accumulated GDDs required to complete the vegetative growth and Flowering stage; MGDDs: the accumulated GDDs for completion of the Maturation stage; HGDDs: the accumulated GDDs for completion of the Harvest stage; $\overline{dEVI_{LTDG\_DL}}$: the mean of daily EVI data reconstructed by *m* groups of EVI observation combinations.

In this study, a lookup table for the FGDDs was established, encompassing a temperature range of 600 to 1200 °C with a sampling interval of 50 °C (i.e., 600, 650, 700, ..., 1200). For each crop pixel, a total of fourteen $dEVI_{LTDG\_DL}$ datasets were generated. These fourteen datasets were then clustered to create the $LEVI_{reference,n}$, denoted as *n* ($n \in 14$). The optimized daily Landsat EVI profile, referred to as $LEVI_{optimized}$, was determined by evaluating its RMSE against Landsat EVI observations on all observation dates, as follows:

$$LEVI_{\text{optimized}} \approx \min(\sum_{j=1}^{n} LEVI_{reference,j}, Landsat_{obs}) \tag{10}$$

The development of the FGDD lookup table adhered to the following criteria: (1) FGDDs > MGDDs > HGDDs; (2) the sum of FGDDs, MGDDs, and HGDDs should be equal to or less than the total accGDDs of the crop growing seasons. Crop growing seasons are defined as the period starting on $DOY_{ini}$ and ending on the first autumn date when the daily air temperature falls below $T_{base}$.

## 3. Algorithm Validations

Validations of the LTDG_DL were conducted using three independent datasets. The first dataset consisted of three experimental fields; the second dataset included agricultural landscapes in the Haean basin; the third dataset encompassed regional agricultural landscapes in Shandong Province. These datasets presented various climate types, different crop types in distinct cropping systems, and varied densities of Landsat EVI observations throughout the crop growing seasons in each agricultural system. The Haean basin and Shandong Province are characterized by fragmented and mosaic agricultural landscapes, which aligned well with the selection of the validation objects pertaining to our research question. Additionally, auxiliary information regarding crop phenology and weather station data was accessible in the Haean basin.

Four variables were used to assess the accuracy of EVI reconstruction with consideration given to the crop growth trajectory. These variables included the EVI increment slope, the EVI decline slope, the absolute EVI, and the timing of peak EVI ($T_{peak\_EVI}$). It should be noted that the $T_{peak\_EVI}$ derived from the crop pixels with a low density of Landsat EVI observations may not accurately represent the field conditions. For the validation of the DL, LTDG, and LTDG_DL methods, crop pixels in the Haean basin that had a peak EVI reasonably close to the flowering stage were selected. To obtain detailed information on crop phenology in the Haean basin, the study by Lindner et al. [3] was consulted. The RMSE and $R^2$ were calculated as indictors of modeling accuracy.

### 3.1. Introduction of the Three Experimental Fields and Parameter Settings of the LTDG_DL

Four crops were selected for algorithm validation in well-managed experimental fields located in Yanhu in China (34°27′03″N, 117°07′11″E), Nebraska in the USA (41°10′46.8″N, 96°26′22.7″W), and Gimje in the Republic of Korea (35°45′N, 126°56′E) [51–53]. The $DOY_{ini}$ settings were determined based on the onset of the consistent increase phase of Landsat EVI observations. Portable multi-spectral sensors were used to extract field EVI measurements from these studies. The EVI increment slope, which represents the increase in the EVI per day during the vegetative growth stage, and the EVI decline slope, which represents the decrease in the EVI per day during the senescence stage, derived from the field EVI measurements were considered as benchmarks to evaluate the performances of the DL, LTDG, and LTDG_DL methods. The calculation modes of the EVI increment and decline slopes are illustrated in the inset of Figure 5h.

The Landsat EVI observations of the growing season were categorized into two scenarios, as shown in Figure 5a–d. The first scenario, labeled S1 manipulation, represented the situation where the EVI observations were intentionally excluded during the periods of rapid green-up and senescence. This is depicted by small triangles. The second scenario, labeled S2 manipulation, depicted the condition where the EVI observations were purposely omitted around the date of the maximum EVI. This is represented by big circles. The S1 and S2 scenarios exemplified the instances of low-density Landsat EVI observation and irregular seasonal distributions, which are commonly observed at larger spatial scales [15,16]. For the validation of the DL, LTDG, and LTDG_DL methods, all Landsat EVI observations during the crop growing seasons were utilized.

The year-round daily meteorological data ($T_{air}$ and $R_s$; see Supporting Information S2) were obtained from the Global Land Data Assimilation System Version 2 products (GLDAS2.0) at the same location and year. These data are available at a daily resolution with a spatial resolution of 0.25° × 0.25° (https://disc.gsfc.nasa.gov/datasets?keywords=GLDAS) (accessed on 5 March 2023). The year-round Landsat 5/7/8 surface reflectance images,

centered on the farming lands, were downloaded from the U.S. Geological Survey (USGS) Earth Explorer on 5 July 2022. These images underwent radiometric and atmospheric corrections using the radCor function of the RStoolbox package in the R language (R Core Team, 2021) [54]. Since Landsat 5 and 7 exhibit a good and consistent match in surface reflectance [55], no further adjustments were necessary regarding sensor geometry. However, the Landsat 5/7 surface reflectance was adjusted to be compatible with Landsat 8 using the band-specific surface reflectance linear regression parameters provided by Roy et al. [56]. Data points with poor Quality Assurance (QA) or blue reflectance of $\geq 0.2$ were considered outliers due to contamination from clouds or aerosols [24]. Reflectance data in the blue ($\rho_{blue}$, 450–520 nm), red ($\rho_{red}$, 630–690 nm), and near-infrared ($\rho_{nir}$, 760–900 nm) bands were used to calculate the EVI [38]. It should be noted that the radCor processing applied to remove cloud and shadow pixels for atmospheric corrections was not perfect [54]. As a result, residual clouds and shadows may appear as distinct outliers in the EVI time series. Due to the irregular observations and low density of observation during the crop growing seasons, reliable outlier detection could not be performed [26]. Therefore, outlier detection techniques were not used to evaluate the radCor-processed Landsat 5/7/8 EVI images.

### 3.2. Introduction of Agricultural Landscapes in Haean Basin and Parameter Settings of LTDG_DL

The Haean basin (128°5′–128°11′E, 38°13′–38°20′N, 297 lines × 385 rows) is located in Yanggu county in Gangwon Province, the Republic of Korea. The topography of the basin resembles a "punch-bowl" shape, with a north–south length of 10.5 km and an east–west width of 6.7 km. Within the Haean basin, a single-cropping system is implemented. Based on field census data from 2009 and 2010 [7], the average parcel area for different crops in the Haean basin is as follows: paddy rice 0.91 ha, potato 0.69 ha, orchard 0.81 ha, bean 0.47 ha, cabbage 0.73 ha, radish 0.69 ha, and ginseng 0.80 ha. The mean parcel area of all croplands is $0.72 \pm 0.13$ ha (mean ± standard deviation). The land use and cover change in the Haean basin during the years 2009 and 2010 are presented in Supporting Information S3.

Field EVI measurements in the Haean basin were not initiated. Surface reflectance images from Landsat 5/7 were downloaded from the Earth Explorer on 26 September 2021 for the years 2009 and 2010. Figure 4 illustrates EVI observational images throughout the crop growing seasons. In 2009, less than 50% of the basin pixels had a fraction of growing season Landsat EVI observations of approximately 91%. In 2010, approximately 87% of the basin pixels had a rate of 61%. A significant proportion had a low fraction of growing season EVI observations (20–40%) in 2009. With the exception of cloud-free data during the early and late growth stages, most pixels had only one or two to three cloud-free EVI observations within the rapid and peak growing seasons.

The landscape-level $DOY_{ini}$ for the onset of vegetation growth (120 DOY) was determined by analyzing the seasonal EVI curves collected from fifty crop pixels. To quantify the landscape-level MGDDs and HGDDs, a meta-analysis of published data on crops grown in the Haean basin was conducted, resulting in values of 300 and 150 °C for the MGDDs and HGDDs, respectively. Two standard weather stations were established in the agricultural lands, one in paddy rice and the other in the dry field zone on an east-facing slope. These stations recorded half-hourly meteorological factors such as incident solar radiation, air temperature, wind speed and direction, relative humidity, and rainfall. Previous research conducted from July to December 2009 revealed insignificant variations in incident solar radiation and daily maximum and minimum air temperatures between the two standard weather stations [3]. Based on these findings, it can be reasonably assumed that all sampled crops in the agricultural landscape developed under similar climate conditions.

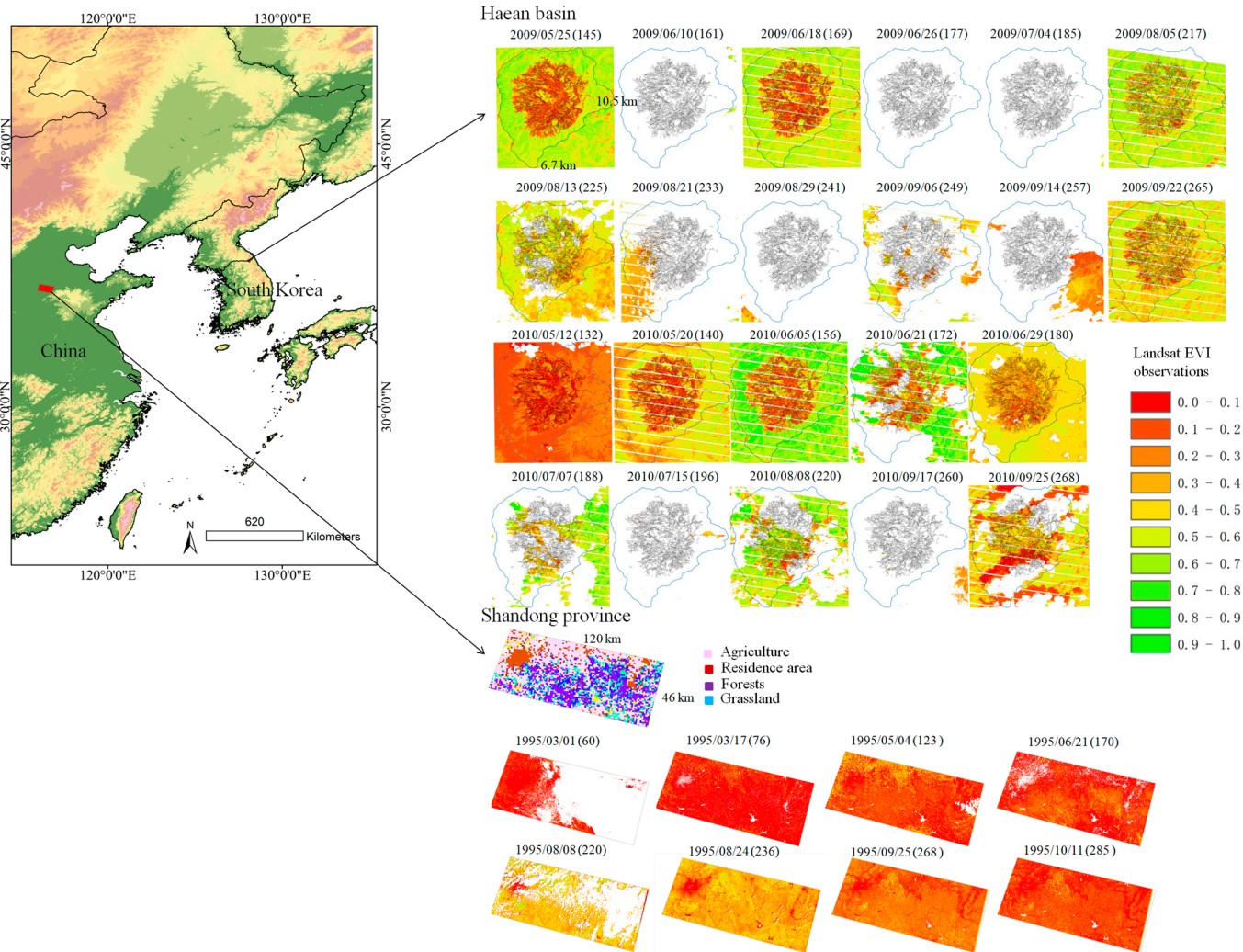

**Figure 4.** Spatial distributions of the Landsat 5/7 EVI observations from May to October in Haean basin in 2019 and 2010 and from March to October in Shandong Province in 1995. The agricultural land use shapefile was stacked on the Landsat 5–7 EVI images in the Haean basin (gray areas). Observation dates are displayed as year/month/day (day of the year). EVI: Enhanced Vegetation Index.

### 3.3. Introduction of Regional Agricultural Landscapes in Shandong Province and Parameter Settings of LTDG_DL

This was a double-cropping system, with a rotation of spring and summer crops, in the regional agricultural landscapes of Shandong Province of China (114°19′–122°43′E, 34°22′–38°23′N). The area covered a length of 120km and a width of 46km, consisting of 3993 lines and 2229 rows. The land use and cover data layer, with Level-2 classification information (i.e., paddy fields and upland fields), at a resolution of 30 m in 1995, was obtained from the Resource and Environmental Science and Data Center (https://www.resdc.cn/) (accessed on 15 December 2022). The crop types present in each pixel were unknown, and field EVI measurements were not available in Shandong Province. Only Landsat 5 TM surface reflectance images were available in 1995. Landsat observational images were classified into four land cover types for the period between March and October in Shandong (Figure 4). For spring crops, only four observational images were available from February to mid-June. Among these, two were acquired at 60 and 76 DOY when crop growth was at a high momentum. Another image was obtained on 170 DOY (the end of June) during the harvest stage of the spring crops. The most-valuable Landsat image was acquired on 123 DOY, corresponding to the flourishing growth stage of the spring

crops. Regarding the summer crops, four Landsat observational images were available. One image was acquired on 220 DOY, capturing the vigorous growth of summer crops, although a significant portion was contaminated by clouds. The observational images acquired on 236, 268, and 285 DOY had minimal cloud contamination, but only the image acquired on 236 DOY was considered valuable due to the likely maximum canopy size of the summer crops. The density of Landsat EVI observations for the spring and summer crops was very low.

The $DOY_{ini}$ values for the spring and summer crops at the landscape level were determined based on the onset of the consistent increase phase of seasonal EVI observations in fifty randomly selected pixels. This approach was consistent with the $DOY_{ini}$ determination in the Haean basin. Prior knowledge of the life cycles of locally prevalent crops in Shandong Province aided in determining the landscape-level $DOY_{ini}$. The landscape-level $DOY_{ini}$ values are displayed in Table 1. The GLDAS2.0 daily meteorological products for the entire year in the same area were resampled to a 30 m resolution.

**Table 1.** Settings of the landscape-level $DOY_{ini}$ in crop pixels in Shandong Province. $DOY_{ini}$: the initial date of seedling emergence/transplanting.

| Site | Crop Types | Life Cycle Classification | Edaphic Classification | Years | $DOY_{ini}$ |
|------|-----------|---------------------------|------------------------|-------|-------------|
| Shandong Province | Paddy or upland crops (unknown crop types) | Unknown | Paddy/upland field | 1995 | 55 for spring crops; 170 for summer crops |

## 4. Results

### 4.1. Performance of the LTDG_DL Algorithm in Experimental Fields

Overestimation of the EVI around the date of peak EVI was observed in the LTDG S1 and S2 simulations (Figure 5a–d). The DL S1 simulation in Nebraska and the DL S1 and S2 simulations in Gimje significantly underestimated the EVI values around the date of the maximum EVI. Satisfactory performances were observed in the LTDG_DL S1 and S2 simulations, with $R^2$ and RMSE values of 0.71 and 0.09 and 0.84 and 0.07, respectively (Figure 5g). The EVI reconstruction accuracy of LTDG_DL S2 was superior to that of LTDG_DL S1, suggesting the importance of having Landsat EVI observations during periods of rapid green-up and senescence for daily EVI reconstruction.

For the S1 scenario, the simulated EVI increment slope decreased from LTDG and LTDG_DL to DL (Figure 5h). There were no significant differences in the EVI increment slope between the field observation and the LTDG simulation. The deviation amplitude of the DL simulation in the EVI increment slope decreased from 74% to 48% in the LTDG_DL simulation. The EVI decline slope simulated by LTDG_DL was 48% lower than that of the DL simulation, while being closer to the field observation. The improved prediction accuracy of the EVI increment slope in the S2 scenario was also achieved by LTDG_DL (Figure 5i).

### 4.2. Performance of the LTDG_DL Algorithm in Agricultural Landscapes of Haean Basin

The Landsat EVI observations and simulations in the Haean basin are displayed in Figure 6, using the LTDG_DL, LTDG, and DL methods. A case study examining paddy rice demonstrated that the $dEVI_{DL}$ preserved the Landsat EVI observations of the rice growing seasons (Figure 6a), with the simulated $T_{peak\_EVI}$ occurring on 245 DOY. The LTDG method, on the other hand, simulated a $T_{peak\_EVI}$ of 210 DOY, which was close to the observed $T_{peak\_EVI}$ of 220 DOY. The LTDG_DL method simulated a $T_{peak\_EVI}$ of 220 DOY, falling between the $dEVI_{LTDG}$ and $dEVI_{DL}$ $T_{peak\_EVI}$ simulations. However, the $dEVI_{LTDG}$ was significantly higher than the $dEVI_{DL}$ at the maturation stage (after 250 DOY). At this stage, the LTDG_DL method effectively captured the EVI's steady declining trend compared to LTDG. The DL method failed to capture Landsat observations during the EVI peak stage in potato (170 DOY) (Figure 6b). The $T_{peak\_EVI}$ values for $dEVI_{DL}$, $dEVI_{LTDG}$, and

dEVI$_{\text{LTDG\_DL}}$ were 210, 175, and 175 DOY, respectively. In 2009 and 2010, the LTDG_DL method achieved better reconstructions of the seasonal EVI tendency and amplitude in crops such as bean, cabbage, radish, orchard, ginseng, and codonopsis (Figure 6c–p).

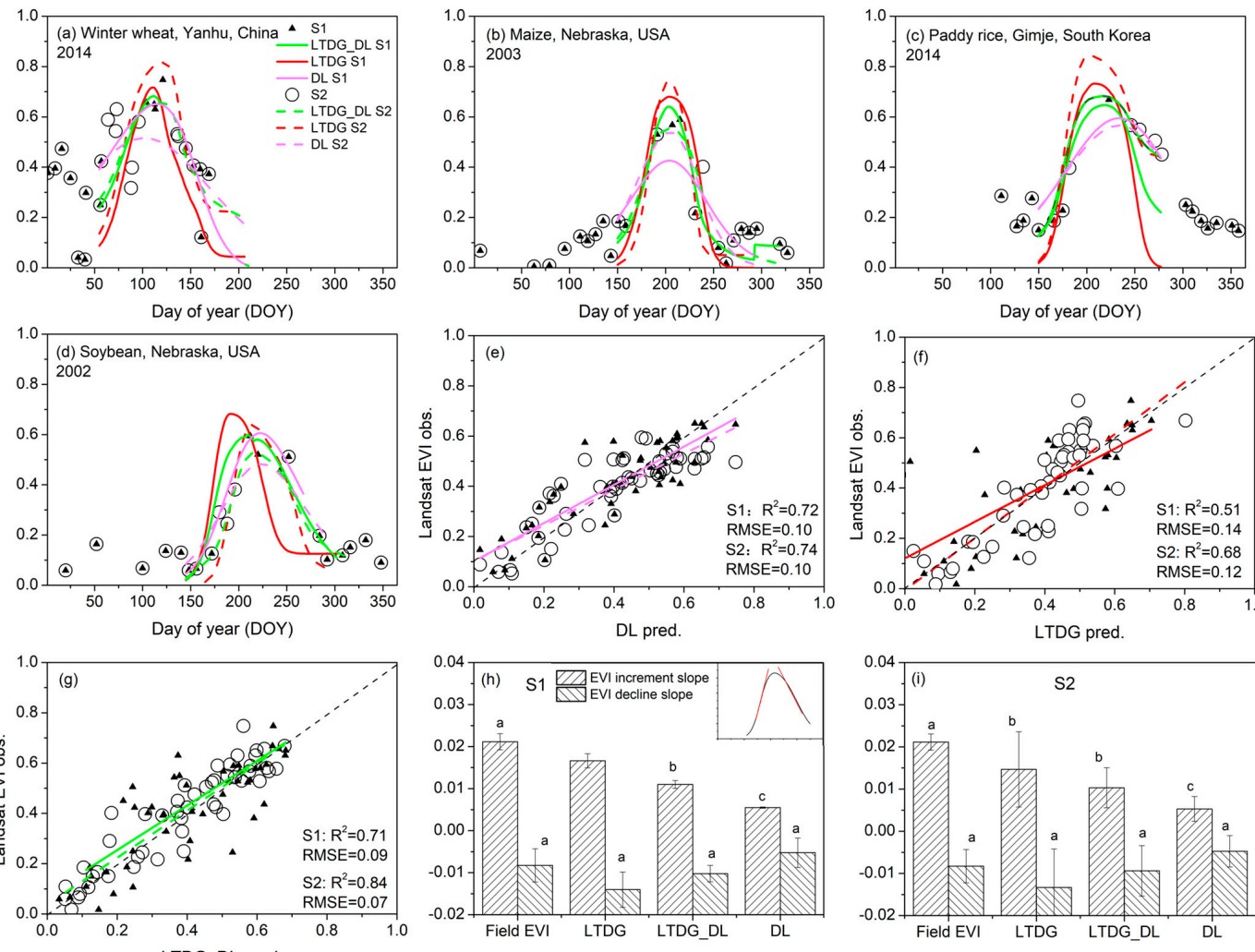

**Figure 5.** Evaluations of the EVI reconstruction accuracy of the DL, LTDG, and LTDG_DL methods in experimental fields in Yanhu, China (**a**), Nebraska in the USA (**b**,**d**), and Gimje in the Republic of Korea (**c**). The S1 manipulation scenario (small triangles) represented the condition of artificially masked EVI observations during the rapid green-up and senescing periods; the S2 manipulation scenario (big circles) represented the case with missing EVI observations around the date of the maximum EVI. All growing season Landsat EVI observations were applied to evaluate algorithm accuracy (**e**,**f**,**g**). Different letters in (**h**,**i**) indicate that, at the 0.05 level, the difference of the population means was significantly different with the null hypothesis (mean1 − mean2 = 0, one-way ANOVA). The 95% error bars were added to the LTDG_DL S1 and S2 simulations. The inset in (**h**) shows the calculation modes of the EVI increment and decline slopes. EVI: Enhanced Vegetation Index; obs.: observation; pred.: prediction; LTDG: the Light- and Temperature-Driven Growth model; DL: the Double-Logistic function; LTDG_DL: the LTDG and DL fusion algorithm.

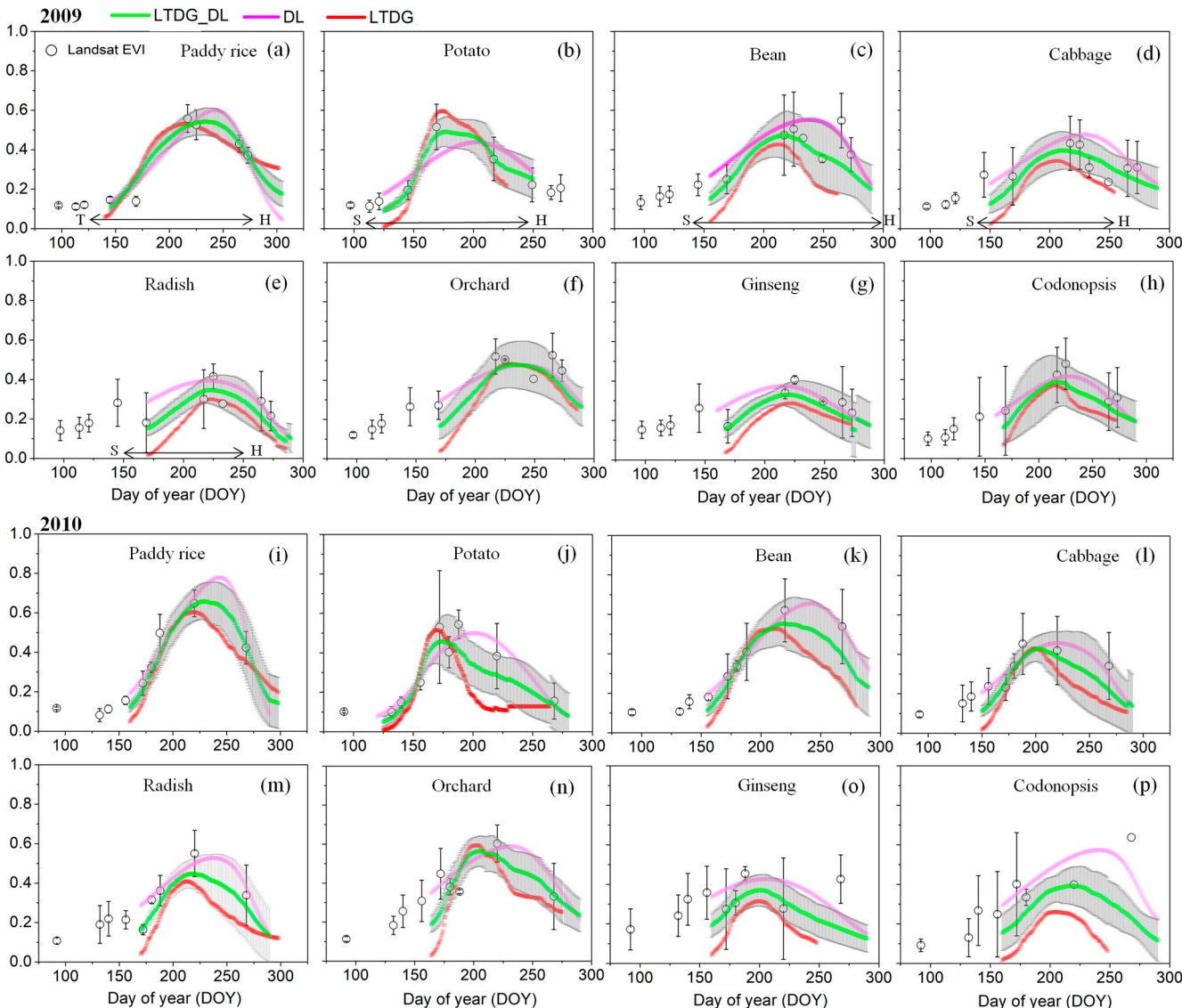

**Figure 6.** Reconstruction of daily Landsat EVI data using the LTDG, DL, and LTDG_DL methods in paddy rice (**a,i**), potato (**b,j**), bean (**c,k**), cabbage (**d,l**), radish (**e,m**), orchard (**f,n**), ginseng (**g,o**), and codonopsis (**h,p**) in the Haean basin. EVI: Enhanced Vegetation Index; T: date of seedling Transplanting; S: Seedling emergence date; H: Harvest date; LTDG: the Light- and Temperature-Driven Growth model; DL: the Double-Logistic function; LTDG_DL: the LTDG and DL fusion algorithm. The 95% error bars indicted by grey areas were added to the LTDG_DL simulations.

The comparisons of Landsat EVI observations and simulations revealed that the RMSE values for DL, LTDG, and LTDG_DL were 0.12, 0.07, and 0.07, respectively (Figure 7a–c). The corresponding $R^2$ values were 0.67, 0.75, and 0.75, respectively. In the $T_{peak\_EVI}$ estimation, the RMSE and $R^2$ values were 4.9 d and 0.53 for DL, 2.68 d and 0.68 for LTDG, and 2.03 d and 0.76 for LTDG_DL (Figure 7d). In 2009, the average EVI increment slope across eight crop types in the LTDG_DL simulations was approximately 0.007, which was 85% higher than that of the DL simulations (Figure 7e), consistent with the findings shown in Figure 5h,i. There were no significant differences in the EVI decline slope between the two methods. Similarly, in 2010, the LTDG_DL yielded an EVI increment slope that was 72% higher than that of the DL method (Figure 7f).

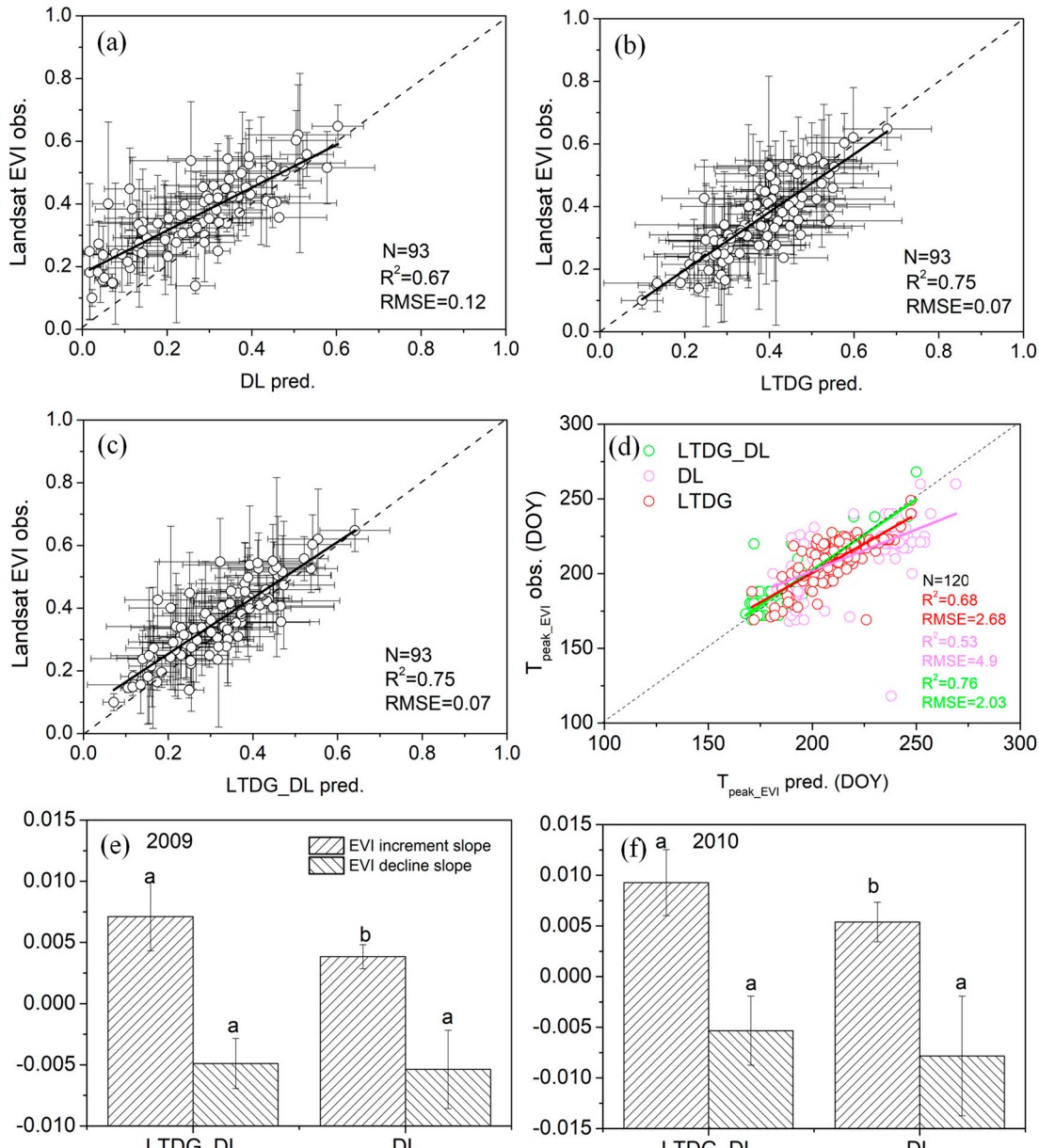

**Figure 7.** Comparisons between Landsat EVI observations and predictions using the DL (**a**), LTDG (**b**), and LTDG_DL (**c**) methods in randomly selected crop pixels in the Haean basin. Evaluations in estimation of $T_{peak\_EVI}$ by the three methods are presented in plot (**d**). Different letters in (**e**,**f**) indicate that, at the 0.05 level, the difference of the population means was significantly different with the null hypothesis (mean1 − mean2 = 0, one-way ANOVA). The dashed line was the 1:1 line (**a**–**c**) and the solid lines were the linear regression fits. LTDG: the Light- and Temperature-Driven Growth model; DL: Double-Logistic function; LTDG_DL: the LTDG and DL fusion algorithm; pred.: prediction; obs.: observation; EVI: Enhanced Vegetative Index; $T_{peak\_EVI}$: the timing of peak EVI.

Figure 8 displays the reconstructed EVI images on the Landsat EVI observation dates in the Haean basin by LTDG_DL. During the period from 145 to 169 DOY in 2009 and 2010, most agricultural lands were dry, while some fields were flooded and prepared for rice seedling transplantation. Seeding emergence did not occur before 160 DOY, causing most agricultural pixels on 132, 140/145, and 156/161 DOY in 2009 and 2010 to appear red (indicating low EVI). On 161 and 169 DOY in 2009, light green pixels were noticeable at the western part of the Haean valley, which are cultivated lands for potato. Potato pixels

became deep green on 185 DOY and returned to light green again on 225 DOY, matching the potato field phenology [3]. Orange dominated the paddy pixels on 177 DOY, then shifted to light yellow on 185 DOY and, finally, turned green on 225 DOY. On 257 DOY, light green and yellow colors indicated senescing canopies. Green colors were observed in upland fields around 217 DOY, which corresponded to the date of maximum canopy growth potential. The LTDG_DL well preserved the spatial details in the EVI time series of various crops in 2010.

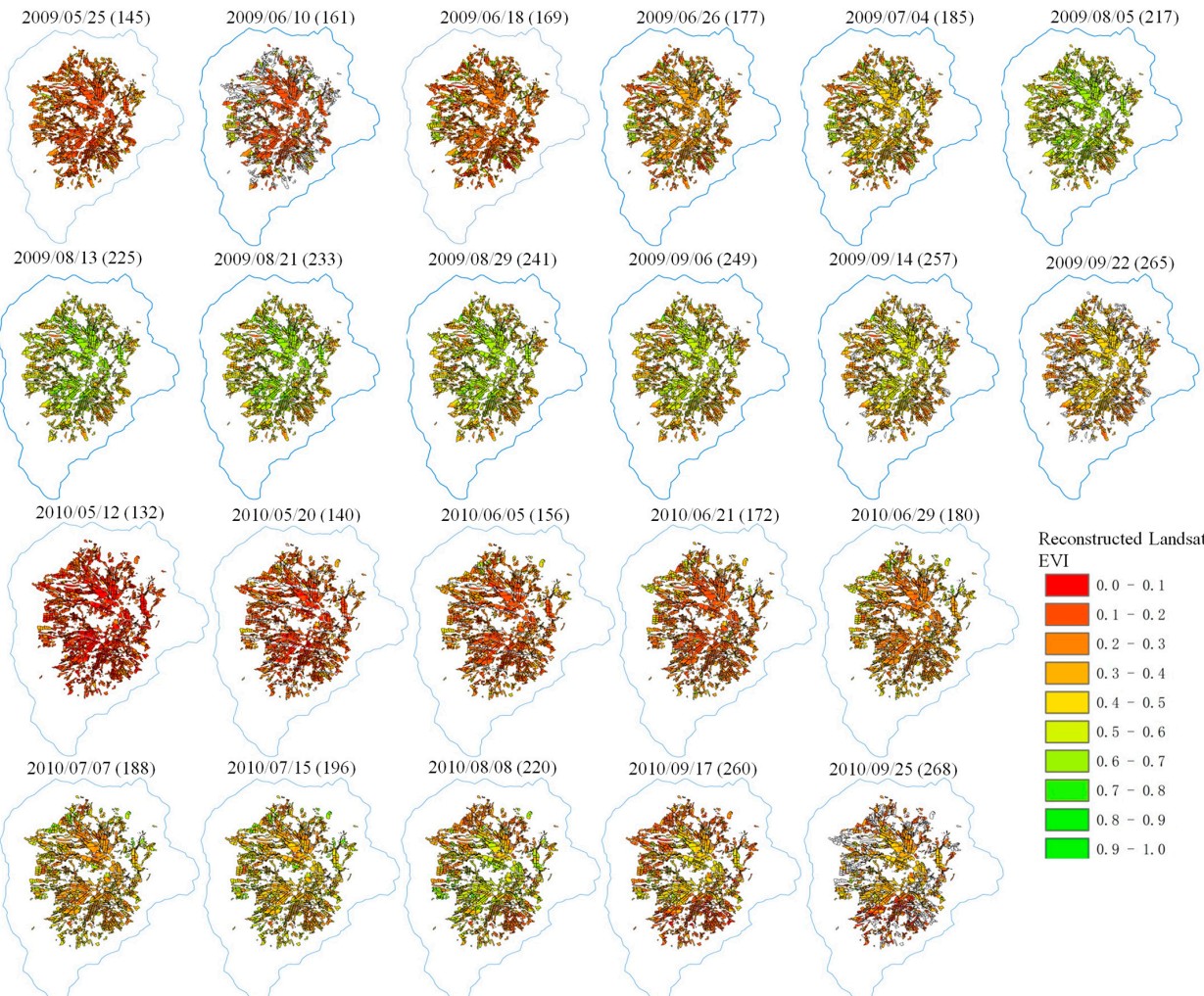

**Figure 8.** Spatial mapping of reconstructed per-pixel EVI on the Landsat EVI observation dates in the Haean basin in 2009 and 2010. The cropland data layer was stacked on the Landsat EVI maps. EVI: Enhanced Vegetation Index; LTDG_DL: the Light- and Temperature-Driven Growth model and the Double-Logistic function fusion algorithm.

### 4.3. Performance of the LTDG_DL Algorithm in Regional Agricultural Landscapes of Shandong Province

Seasonal trajectories of the reconstructed Landsat EVI in four representative fields in Shandong are displayed in Figure 9a–d. The DL method successfully captured the Landsat EVI observations during the spring and summer seasons, demonstrating a high $R^2$ of 0.92 and a low RMSE of 0.05, as depicted in Figure 9e. Similar results were obtained with the LTDG_DL method, as shown in Figure 9g. However, the prediction accuracy declined when using the LTDG simulations, as evidenced in Figure 9f. For the summer crops, the EVI increment slope for the LTDG_DL method was approximately 0.008, while it was around 0.006 for the DL simulation, as indicated in Figure 9h. Statistical analysis revealed significant

differences in the EVI increment slope between the LTDG_DL and DL methods, consistent with the findings in Figures 5h,i and 7e,f. Notably, the LTDG_DL simulation exhibited a higher value in the EVI increment slope of approximately 57% than the DL simulation of the spring crops, as illustrated in Figure 9i. However, no significant differences were observed in the EVI decline slope between the two methods in either the spring or summer crops. In comparison to the DL method, the LTDG_DL method accurately captured the growth pattern of the EVI during the vegetative stage of the spring/summer crops and produced consistent seasonal EVI trajectories (Figure 9a–d). This can be attributed to the superior performance of the LTDG_DL method in estimating the EVI increment slope and its ability to combine the advantages of the DL and LTDG methods.

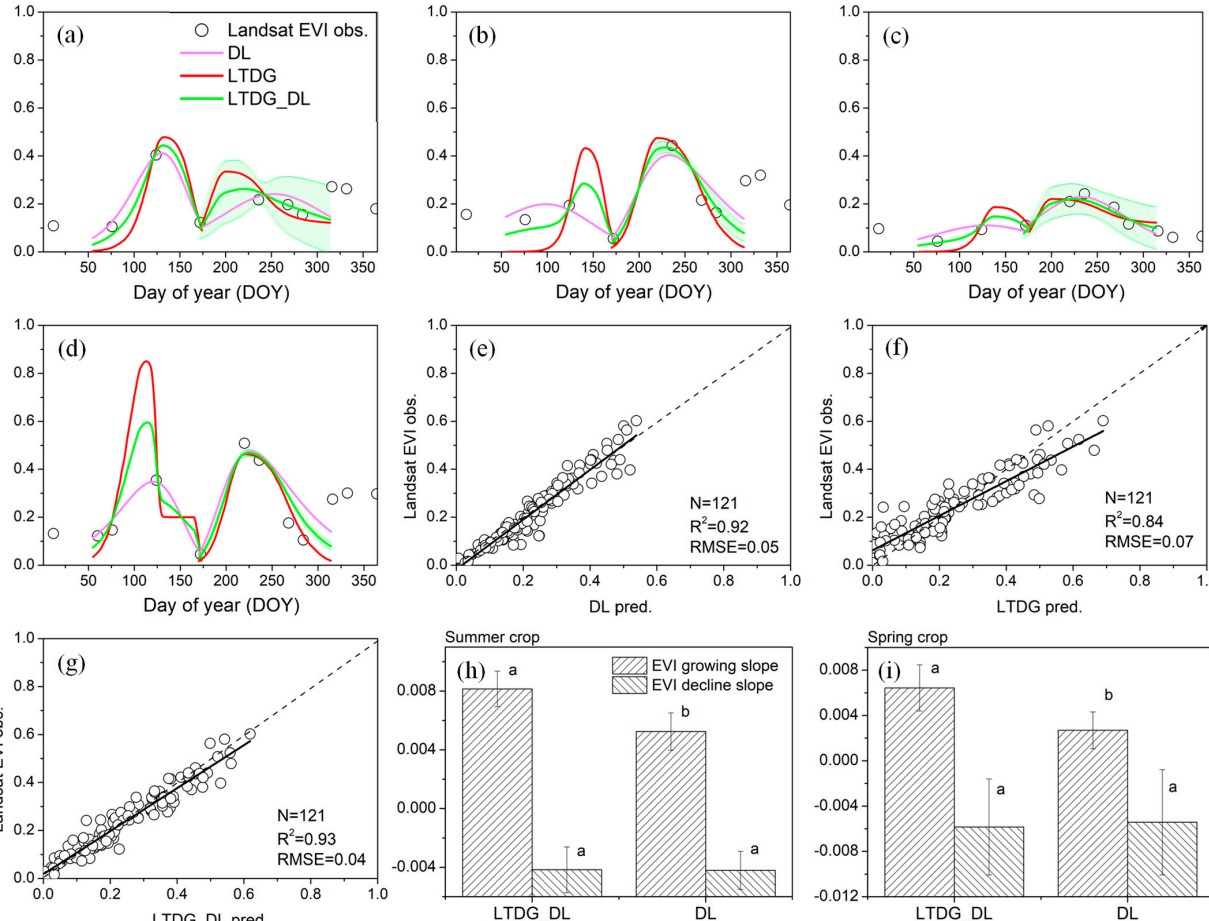

**Figure 9.** Comparisons in the reconstruction accuracy of daily Landsat EVI data among the DL, LTDG, and LTDG_DL methods in randomly selected crop pixels (**a**–**d**) in Shandong Province. Comparisons in EVI observation and predictions by the three methods are presented in plots (**e**–**g**). Different letters in (**h**,**i**) indicate that, at the 0.05 level, the difference of the population means was significantly different with the null hypothesis (mean1 − mean2 = 0, one-way ANOVA). The dashed line was the 1:1 line (**e**–**g**) and the solid lines were the linear regression fits. LTDG: the Light- and Temperature-Driven Growth model; DL: Double-Logistic function; LTDG_DL: the LTDG and DL fusion algorithm; pred.: prediction; obs.: observation; EVI: Enhanced Vegetative Index.

The results of the LTDG_DL EVI reconstruction in Shandong, where crops were grown under various environmental conditions including rainfed, irrigation, fertilization, and no fertilization, yielded a relatively small RMSE of 0.03 ± 0.02 (mean ± SD) for the spring crops (reversed funnel boxplot in Figure 10) and an RMSE of 0.06 ± 0.02 for the summer crops (small waist gourd boxplot in Figure 10). Approximately 75% of the spring crop pixels had an RMSE less than 0.05. The RMSE range of the summer crops exhibited a

distribution shaped like a small-waisted gourd, with 50% of crop pixels having an RMSE less than 0.05 and 25% of pixels having an RMSE ranging from 0.05 to 0.09. The relatively large RMSE in the 25% of pixels was primarily due to the abnormal EVI observational values (approximately 0.01) on 220 DOY of those pixels when the crops grew vigorously. The abnormal low EVI values were not considered outliers and were not removed during the process of calculating the pixel RMSE. LTDG_DL effectively captured the crop seasonal EVI trajectories and spatial features in the double-cropping systems (Figure 10).

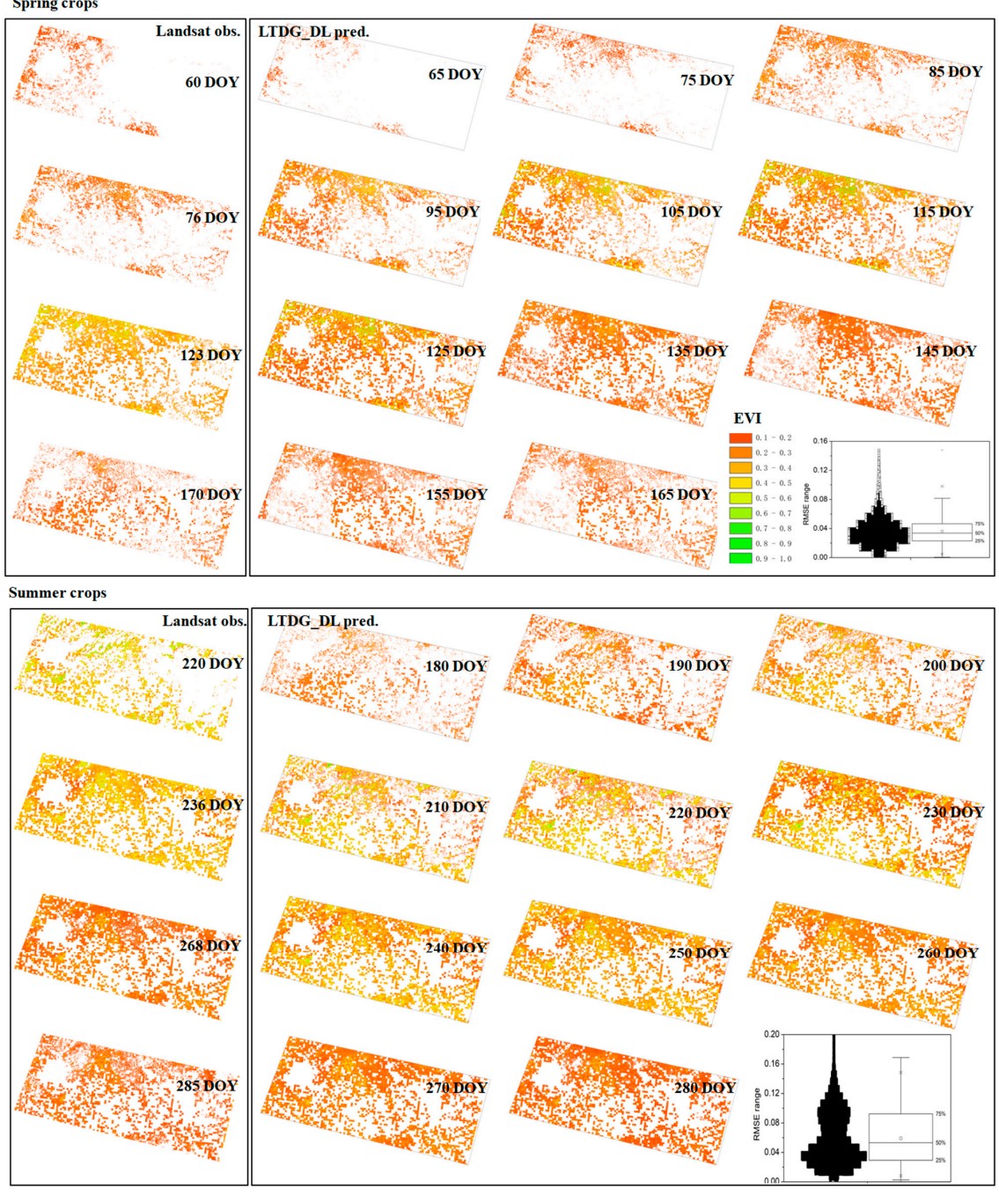

**Figure 10.** Spatial mapping of daily Landsat EVI reconstructed by LTDG_DL in the spring crop + summer crop rotation systems in Shandong Province. Reconstructed Landsat EVI maps are displayed at 10 d intervals. EVI: Enhanced Vegetation Index; LTDG: the Light- and Temperature-Driven Growth model; DL: the Double-Logistic function; LTDG_DL: the LTDG and DL fusion algorithm; DOY: Day Of Year; RMSE: Root-Mean-Squared Error; obs.: observation; pred.: prediction.

## 5. Discussion

The predictive capability of the DL model varied depending on the quantity and temporal distribution patterns of the Landsat EVI. The DL model performed poorly in protecting key Landsat EVI observations in experimental fields with irregular temporal distributions. The DL model was sensitive to noise data in the Landsat EVI observations, confirming the previous findings by Liu et al. [28]. The EVI slope during the rapid expansion of the crop canopies in the vegetative growth stage was significantly lower than the expected conditions, indicating a mismatch between the DL-reconstructed Landsat EVI time series and the crop growth trajectory.

The GF-SG is widely recognized for its superior performance in reconstructing the Landsat EVI compared to previous Landsat–MODIS fusion algorithms [32]. Our study revealed that a significant proportion of agricultural pixels, ranging from 10% to 45%, were identified as invalid due to low $Cor_{j\_(x,y)}$ values (see Figure S4a,b in the Supporting Information S4). This issue is particularly prevalent in highly fragmented landscapes, where there is spatiotemporal incompatibility between the EVI time series of the targeted Landsat pixel and neighboring re-sampled MODIS pixels. The spectral response of the near-infrared wavelength band, which varies across different landscapes, was identified as a critical factor influencing MODIS–Landsat spatiotemporal comparability [57]. Recently, Sun et al. [5] proposed a crop-type-based Crop Reference Curve (CRC) approach, wherein they extracted NDVI time series from MODIS 250 m pure crop pixels identified using the Cropland Data Layer (CDL), and used these data to fit Landsat-like observations. Specifically, they selected MODIS candidate curves within a given crop type to match the Landsat observations in pixels where the same crop type was grown. However, it is important to note that land use and land cover data layers in most countries and regions do not accurately register the spatial distributions of crop types. Additionally, the Landsat–MODIS fusion algorithms cannot be applied in years and areas where MODIS products are unavailable. Therefore, there is a pressing need to develop a new algorithm that can address these limitations and enhance the reliability of reconstructed daily Landsat EVI data in mosaic and fragmented agricultural landscapes.

### 5.1. Superiority of the LTDG_DL Algorithm

The LTDG_DL algorithm differs from the Landsat–MODIS spatiotemporal fusion methods. These fusion algorithms are commonly employed to extract valuable auxiliary information from MODIS data and to align MODIS time series with Landsat observations [5,32]. The primary auxiliary information utilized by LTDG_DL is the daily ground meteorological factors $R_s$ and $T_{air}$. Global gridded daily data for $R_s$ and $T_{air}$ throughout the year can be freely obtained from satellite-derived products such as GLDAS and the Modern-Era Retrospective analysis for Research and Applications (MERRA) datasets. The foundation of the LTDG_DL algorithm is based on the biophysical mechanisms of growth driven by light and temperature [42,46], which comprehensively consider climatic factors that influence crop growth and development trajectory on a daily basis. It was assumed that changes in the Landsat EVI during different seasons were related the growth of the crop canopy, which is strongly regulated by ambient light, temperature, and soil water content, as sensed by the plants. This assumption is supported by the close correlation between the LAI and EVI, with the EVI capturing more than 80% of the seasonal variations in the LAI [43,45]. Numerous studies have been conducted utilizing VI time series curves derived from remotely sensed data for detecting crop phenology. Satellite EVI time series curves have been found to be closely linked to crop phenology [58], with small RSME values ranging from 1.6 (silking stage) to 5.6 (dent stage) d for maize and from 2.5 (beginning of maturity) to 5.3 (beginning of seedling emergence) d for soybean [59], 10.0 d in the estimation of the paddy rice establishment date in Italy, India, and the Philippine [60], 8.1–14.7 d in the estimation of the Start of Season (SoS) in single rice, early rice, late rice, spring wheat, and summer maize, and 7.9–13.9 d in the estimation of the End of Season (EoS) among those crops [61]. The start of the growing season identified by the satellite EVI time series

typically lagged behind the actual conditions by approximately 10 d, while there was a notable correspondence in the timing of the flowering stage between the satellite results and field observations. This suggests that the EVI growing slope derived from the satellite EVI time series should be sufficiently high in order to obtain better estimations of the timing of the flowering stage. The algorithm validations through field experiments, as well as in the Haean basin and Shandong Province, indicated that the LTDG_DL algorithm outperformed in predicting the EVI growth slope and $T_{peak\_EVI}$ and safeguarding the Landsat EVI observations. The results demonstrated that the LTDG_DL algorithm, integrating the advantages of a process-based crop model and Landsat EVI observational data, effectively reconstructed the daily Landsat EVI trajectory to align with the crop growth trajectory as closely as possible.

Roy and Yan [26] conducted a study in the Great Plains of North America, analyzing Landsat NDVI data with a rich temporal information (6–9 observations over major crop growing seasons). They found that harmonic models were able to fit the annual time series with RMSE values ranging from 0.05 to 0.08. In another study focusing on the Delmarva Peninsula of the USA, where daily Landsat NDVI reconstruction was performed using 5–9 Landsat observations over major crop growing seasons, the RMSE of the CRC method was found to be 0.07 [5]. However, when attempting to fit coarse Landsat NDVI observations with fewer than three observations within ±20 d using the harmonic modeling method, the resulting RMSE was significantly larger, equal to or greater than 0.12 [27]. The performance of the LTDG_DL algorithm in capturing the seasonal evolution of various crops in the Haean basin was evaluated, and its RMSE value was determined to be 0.07 (Figure 7c). The algorithm successfully represented the non-symmetrical EVI waveform characteristics of paddy rice, winter/spring wheat, maize, and other upland crops (Figure 6). On the other hand, the GF-SG failed to rebuild daily Landsat EVI data in 10–45% of agricultural pixels in complex landscape areas. However, LTDG_DL was able to recreate the daily Landsat EVI profile in those pixels (Figures S4 and S5 in Supporting Information S4). Furthermore, the LTDG_DL algorithm exhibited high accuracy in reconstructing the spring crop EVI in Shandong Province in 1995, as indicated by an RMSE of 0.03. The algorithm proved to be robust against noise and irregular Landsat EVI observations over time, enabling the modeling of rapid canopy changes during the vegetative growth and maturation stages (acceleration/deceleration). Additionally, The LTDG_DL algorithm demonstrated its capability to reconstruct daily Landsat EVI data with relatively high accuracy in crop pixels where crop types were unknown and the Landsat EVI observation density was low at regional scales. To enhance the reconstruction of Landsat data over cloud-prone, fragmented, and mosaic landscapes, as well as for the years prior to 2000, LTDG_DL can utilize auxiliary information from GLDAS2.0 meteorological data and GLC_FCS30 1984–2019 global annual 30 m land use classification products [62]. This integration is expected to improve the accuracy of the reconstructed Landsat EVI data.

### 5.2. Limitations of the LTDG_DL Algorithm

The LTDG_DL algorithm, constrained by Landsat EVI observations, exhibited imperfections. It was concerning that, although our approach had advantages in reconstructing daily Landsat EVI data, it also had some potential drawbacks. The LTDG_DL algorithm demonstrated high modeling accuracy in rainfed fields in Shandong in 1995, capturing the constrained EVI trajectory over a prolonged period of stress. Crop growth, affected by various environmental factors, exhibits inconsistent responses among different crop types [63]. For instance, the current LTDG_DL algorithm is unlikely to capture sudden changes in crop status (such as a heatwave). Short-term stress typically has the most-significant impact on crop physiology, leading to a depression in canopy reflectance signals in the red waveband [64], which subsequently affects satellite EVI observations. Crops experiencing short-term heat stress usually revert to their initial physiological states once the stress is alleviated [65]. In situations where the Landsat EVI observation density is extremely low,

as indicated in Figure 9d, and EVI observations are absent during the flowering stage, LTDG_DL without considering soil water content and drought effects may overestimate the EVI if drought impacts are substantial. Moreover, the current LTDG_DL algorithm does not account for radiation transfer mechanisms, thus failing to capture non-biological singles in canopy reflectance in the red and near-infrared wavebands due to seasonal variations in the solar zenith angle.

Determining the landscape-level $DOY_{ini}$ was the only parameter that needed to be determined. The landscape-level $DOY_{ini}$ can be defined using either the statistical plant phenology extraction method or the endmember EVI ensemble combined with prior knowledge method. Previous studies have shown that the seasonal onset, which is equivalent to $DOY_{ini}$, is often defined as the date when the relatively flat and low EVI phase intersects with the continuously increasing phase of Landsat EVI observations [21,60,66]. The landscape-level $DOY_{ini}$ setting can be determined in crop pixels by identifying the first date of the minimum EVI during a continuously increasing phase of Landsat EVI observations. However, accurately determining $DOY_{ini}$ using the statistical phenology extraction method was not possible in pixels with a very low density of Landsat EVI observations. The combination of the endmember EVI library and the prior knowledge method assists in visually interpreting the landscape-level $DOY_{ini}$ based on the beginning of the consistent increase phase in the seasonal EVI curves obtained from various crop pixels across a research zone (Figure S1 in Supporting Information S1). The seasonal EVI observations obtained from crop pixels were combined to create an endmember EVI library. Knowledge of the crop growth history of predominant local crops aided in determining the landscape-level $DOY_{ini}$. The improved reconstruction of Landsat EVI data in Shandong Province and the Haean basin indicated that the semi-empirical determination of the landscape-level $DOY_{ini}$ using the seedling emergence date of local prevailing crops was highly beneficial. The local seeding calendar within the specific agroclimatic zone was relatively consistent over long periods, as indicated by [67,68]. Determining the climate zone suitable for applying the LTDG_DL method involves a balance between zonal size and the consistency of the local seedling emergence calendar, which is more feasible at the state/province level where agroclimatic conditions are comparable.

## 6. Conclusions

Agricultural ecosystems play a critical role in regulating seasonal variations in carbon and water fluxes at regional and global scales [69]. The EVI time series data are crucial input parameters in numerous land surface models developed to simulate vegetation carbon and water fluxes between the land surface and the atmosphere. Therefore, it is necessary to improve the reliability of Landsat EVI reconstruction for intensive agricultural landscapes. In this study, we developed a parameter-constrained LTDG_DL algorithm and made evaluations in a large range of agricultural systems associated with planting culture, crop types, and field management. The results suggested that LTDG_DL was superior in the predictions of the EVI growing slope and the timing of peak EVI and in the protection of key points in crop pixels with irregular Landsat EVI observations over time. The parameterization of LTDG_DL can be readily set up at a regional scale.

**Supplementary Materials:** The following Supporting Information can be downloaded at: https://www.mdpi.com/article/10.3390/rs15194673/s1, Figure S1. The $DOY_{ini}$ setting by empirically loading the onset of the consistent increase phase of Landsat EVI observations in crop pixels. $DOY_{ini}$: the initial date of seedling emergence/transplanting; obs.: observation. Figure S2. Daily incident solar radiation and air temperature profiles in Yanhu, China, in 2014, Gimje in the Republic of Korea in 2014, and Nebraska in the USA in 2002 and 2003. Data were extracted from the Global Land Data Assimilation System Version 2 (GLDAS2.0). Figure S3. Land use and land change in the Haean basin in 2009 and 2010. Figure S4. Accumulated percentage of invalid pixels that cannot be identified by the GF-SG in eight crops and other upland crops in the Haean basin in 2009 and 2010 (Subplots a and b). Subplots c and d show comparisons of the reconstructed daily Landsat EVI data by the LTDG_DL and GF-SG algorithms for potato and cabbage pixels. Two potato and cabbage pixels that

can be detected as valid pixels using the GF-SG algorithm were used in Subplots c and d. $\text{Cor}_{j\_(x,y)}$: correlation coefficient between the Landsat pixel (x, y) and the neighboring pixels in the local window of 40 × 40 px centered at pixel (x, y) (j = 1:1600) (Chen et al., 2021 [32]). EVI: Enhanced Vegetative Index; LTDG: the Light- and Temperature-Driven Growth model; DL: the Double-Logistic function; LTDG_DL: the LTDG and DL fusing algorithm; GF-SG: the Gap-Filling and Savitzky–Golay filtering method. Figure S5. Daily Landsat EVI data reconstructed by the LTDG_DL algorithm for crop pixels where applications of the GF-SG were hindered in the Haean basin. $\text{Cor}_{j\_(x,y)}$: correlation coefficient between the Landsat pixel (x, y) and the neighboring pixels in a local window of 40 × 40 px centered at pixel (x, y). EVI: Enhanced Vegetative Index; LTDG: the Light- and Temperature-Driven Growth model; DL: the Double-Logistic function; LTDG_DL: the LTDG and DL fusion algorithm; GF-SG: the Gap-Filling and Savitzky–Golay filtering method.

**Author Contributions:** Conceptualization and investigation, methodology, supervision, visualization, and writing—original draft: W.X.; writing—review and editing: J.K., R.C., Z.Y. and W.X. All authors have read and agreed to the published version of the manuscript.

**Funding:** This work was financially supported by the National Natural Science Foundation of China (32001129) and the Open Funding from CAS Key Laboratory of Tropical Forest Ecology (22-CAS-TFE-03).

**Data Availability Statement:** Code script of the LTDG_DL algorithm is available in the online materials.

**Conflicts of Interest:** The authors declare no conflict of interest.

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
