# Peer review of "Improving Reliability in Reconstruction of Landsat EVI Seasonal Trajectory over Cloud-Prone, Fragmented, and Mosaic Agricultural Landscapes"

_remotesensing, doi:10.3390/rs15194673_

Round 1

Reviewer 1 Report

This manuscript developed a LTDG_DL algorithm, which can achieve reconstruction of the Landsat 5/7 EVI seasonal trajectory in cloud-prone and fragmented landscape-complex agricultural regions. The experiments were conducted in different regions and the results are promising. However, all the figures are so unclear, which may hinder the reading. In general, this study is innovative; here are some further suggestions:

1 All the figures in the manuscript and SI are not clear. It is hard to understand the information from the current version. Please must provide high resolution figures.

2 Since Landsat EVI and EVI reconstruction are keywords, what is the significance and necessity to improve the reliability in reconstruction of Landsat EVI? Why you focus on EVI instead of other vegetations index? The current introduction is well written, but most contents are related to the methods. It's important for the authors to provide a clear justification for focusing on EVI and why this improvement is important in the context of their research

3 Line 301, 325, 327, unclear message here.

4 Study areas in this research should be presented. Such information like geographic locations, characteristics, and relevance to the research should be clarified.

5 It seems the experiments in Shandong and Haean basin was conducted in different years. How you choose the year? And is the model robust when used in other years? It's essential for the authors to explain their rationale for selecting specific years and to conduct robustness analysis if applicable.

6 What’s the further application for the Landsat EVI in future studies? Authors can discuss potential applications, implications, and areas where Landsat EVI data can be beneficial for future research.

English should be polished, especially for the figures.

Author Response

Comments and Suggestions for Authors

This manuscript developed a LTDG_DL algorithm, which can achieve reconstruction of the Landsat 5/7 EVI seasonal trajectory in cloud-prone and fragmented landscape-complex agricultural regions. The experiments were conducted in different regions and the results are promising. However, all the figures are so unclear, which may hinder the reading. In general, this study is innovative; here are some further suggestions:

All the figures in the manuscript and SI are not clear. It is hard to understand the information from the current version. Please must provide high resolution figures.

Reply: Thanks for your suggestions. All figures involved in the revised manuscript have been updated using high resolution versions. Furthermore, we added geophysical map of Haean basin and Shandong province in figure 4 where Landsat EVI reconstruction experiments were carried out. Please kindly check them.

Since Landsat EVI and EVI reconstruction are keywords, what is the significance and necessity to improve the reliability in reconstruction of Landsat EVI? Why you focus on EVI instead of other vegetations index? The current introduction is well written, but most contents are related to the methods. It's important for the authors to provide a clear justification for focusing on EVI and why this improvement is important in the context of their research

Reply: Agricultural ecosystems play a critical role in regulating seasonal variations in carbon and water fluxes at regional and global scales (Zeng et al., 2014). This is due to their increased carbon uptake capacity, particularly during summer seasons, as indicated by the leaf area index (Haung et al., 2018). Our study revealed that the accuracy of reconstructing the seasonal trajectory of crop EVI has a significant impact on estimating the daily evapotranspiration rate (Xue and Ko, 2021). Therefore, it is necessary to improve the reliability of Landsat EVI reconstruction.

The reason why we focus on EVI lies on two reasons. One is that the EVI that can reduce the saturating effects of dense canopy as much as possible performs better than NDVI (Huete et al., 2002). The other is that the LTDG_DL algorithm is an extension version of the one proposed by our previous research (Xue and Ko, 2021) that also selects EVI as the target.

The reason why we focus on EVI was partially highlighted in the third paragraph of introduction part. The ecological background regards on reconstruction accuracy of crop EVI seasonal trajectory has substantial impacts on estimation of daily evapotranspiration rate was not supplemented in the first paragraph of introduction part, because we try to limit the length of the introduction part and make it more concise.

Reference

Zeng, N., et al., 2014. Agricultural Green Revolution as a driver of increasing atmospheric CO2 seasonal amplitude. Nature, 515: 394–397.

Huang, K., et al., 2018. Enhanced peak growth of global vegetation and its key mechnisms. Nature Ecology Evolution, 2: 1897–1905.

Xue, W., Ko, J., 2022. Radiation estimation and crop growth trajectory reconstruction by novel algorithms improve MOD16 evapotranspiration predictability for global multi-site paddy rice ecosystems. Journal of Hydrology, 612, 128204.

Huete, A., et al., 2002. Overview of the radiometric and biophysical performance of the MODIS vegetation indices. Remote Sensing of Environment, 83: 195–213.

3 Line 301, 325, 327, unclear message here.

Reply: We invited a native English speaker to polish the manuscript. We highlighted all changes with marked labels.

4 Study areas in this research should be presented. Such information like geographic locations, characteristics, and relevance to the research should be clarified.

Reply: Yes, we agree. We added geophysical map of Haean basin and Shandong province in figure 4 where Landsat EVI reconstruction experiments were carried out. Please kindly check them.

5 It seems the experiments in Shandong and Haean basin was conducted in different years. How you choose the year? And is the model robust when used in other years? It's essential for the authors to explain their rationale for selecting specific years and to conduct robustness analysis if applicable.

Reply: The selection of the year for model validation was based on the percentage of high-quality Landsat EVI observations during crop growing seasons. It should be noted that there was no high-quality Landsat data available during the peak growth stage of the spring crop in Shandong in 1995, and only a limited number of high-quality Landsat EVI observations were available during crop growing seasons. It was an ideal site to conduct the validation of the LTDG_DL algorithm.

6 What’s the further application for the Landsat EVI in future studies? Authors can discuss potential applications, implications, and areas where Landsat EVI data can be beneficial for future research.

Reply: The EVI time series data are crucial input parameters in numerous land surface models developed to simulate vegetation carbon and water fluxes between the land surface and the atmosphere. Regardless of whether we decide to extend the length of the manuscript, we have allocated space to discuss the potential applications of the Landsat EVI data in the conclusion part.

 Comments on the Quality of English Language: English should be polished, especially for the figures.

Reply: We invited a native English speaker to polish the manuscript. We highlighted all changes with marked labels.

Reviewer 2 Report

To address the question of how to reduce uncertainty in Landsat EVI daily series reconstruction, a novel parameter constrained Light and Temperature-Driven Growth (LTDG) and Double Logistic (DL) fusion algorithm (abbreviated as LTDG_DL) was developed in this study. Overall, the efforts made herein are of great reference for Landsat VI reconstruction. But the manuscript still needs some revisions before it can be accepted.

1. The image resolution in the text is too low;

2. The conclusion section is too concise.

Author Response

Comments and Suggestions for Authors

To address the question of how to reduce uncertainty in Landsat EVI daily series reconstruction, a novel parameter constrained Light and Temperature-Driven Growth (LTDG) and Double Logistic (DL) fusion algorithm (abbreviated as LTDG_DL) was developed in this study. Overall, the efforts made herein are of great reference for Landsat VI reconstruction. But the manuscript still needs some revisions before it can be accepted.

  1. The image resolution in the text is too low;

Reply: Thanks for your suggestions. All figures involved in the revised manuscript have been updated using high resolution versions. Furthermore, we added geophysical map of Haean basin and Shandong province in figure 4 where Landsat EVI reconstruction experiments were carried out. Please kindly check them.

  1. The conclusion section is too concise.

Reply: Revisions in the conclusion section have been made. Please kindly check them.

Reviewer 3 Report

The manuscript aims to retrieve high temporal resolution vegetation index. The manuscript is of interest to the readers. I have some minor queries about the manuscript.

1) The usage of fixed values of various constants is an issue. examples, in equation (1), the ranges of values fixed to x2 and x4, in equation (2), the fixed value of LUE etc. x2 and x4 depends on season and will vary geographically. Similarly, LUE will vary for each crop. Why these fixed values are used? They can affect the results significantly.

2) Similarly, how the coefficients in equation 8 was fixed? For which crop it is valid?

3) Generally, it will be extremely difficult to know information about cropping seasons and phenology over a heterogeneous cropping region as there will be multiple crops each having different crop calendar. Same crop may have different sowing dates affecting the phenology over the region. This will be an issue over most of the south asian countries.  Since this information is critical for the model calibration, this will affect the accuracy of the estimated EVI. How to overcome this issue?

4) The methodology section and the validation was little difficult to understand for me and it required multiple reading. Please rewrite the text to make it simpler to the reader.

5) The figures are unreadable. Similarly, some notations (e.g. line 301 in page 10) embedded as figures within text are not clear. Make them readable and use high resolution figures.

As mentioned before, please simplify the methodology section and the results section for easy understanding.

Author Response

Comments and Suggestions for Authors

The manuscript aims to retrieve high temporal resolution vegetation index. The manuscript is of interest to the readers. I have some minor queries about the manuscript.

1) The usage of fixed values of various constants is an issue. examples, in equation (1), the ranges of values fixed to x2 and x4, in equation (2), the fixed value of LUE etc. x2 and x4 depends on season and will vary geographically. Similarly, LUE will vary for each crop. Why these fixed values are used? They can affect the results significantly.

Reply: There are many parameters involved in the LTDG_DL algorithm, and some of them can be constrained within a narrow range such as x2 and x4 in equation (1). x1 and x3 are the locations of the left (the increasing phase) and right (the decreasing phase) inflection points, respectively; x2 and x4 determine the rates of changes at these points. The DL model, which incorporates constrained values for x2 and x4, demonstrates the ability to achieve satisfactory accuracy in predicting EVI time series data for temperate crops (Xue et al., 2021). Note that the range of x2 and x4 may vary between temperate and tropical crops. This has been highlighted in the revised manuscript. We agree with you that there may significant changes in LUE along crop growing seasons. In future study, we may incorporate the LUE products retrieved from satellite products to support our algorithm.

Reference

Wei Xue, Seungtaek Jeong, Jonghan Ko, Jong-Min Yeom. Contribution of biophysical factors to regional variations of evapotranspiration and seasonal cooling effects in paddy rice in South Korea. Remote Sensing, 2021, 1374779.

2) Similarly, how the coefficients in equation 8 was fixed? For which crop it is valid?

Reply: Achieving a balance between model complexity and model accuracy necessitates making certain assumptions about model parameters. Previous studies (Xue and Ko, 2022; Shawon et al., 2020) have shown that the relationship between EVI and LAI in crops adheres to a general experimental curve. This relationship may also hold true for graminaceous crops.

Reference

Xue, W., Ko, J., 2022. Radiation estimation and crop growth trajectory reconstruction by novel algorithms improve MOD16 evapotranspiration predictability for global multi-site paddy rice ecosystems. Journal of Hydrology, 612, 128204.

Shawon et al., 2020. Assessment of a proximal sensing-integrated crop model for simulation of soybean growth and yield. Remote Sensing, 12, 410.

3) Generally, it will be extremely difficult to know information about cropping seasons and phenology over a heterogeneous cropping region as there will be multiple crops each having different crop calendar. Same crop may have different sowing dates affecting the phenology over the region. This will be an issue over most of the south asian countries.  Since this information is critical for the model calibration, this will affect the accuracy of the estimated EVI. How to overcome this issue?

Reply: Recurring Landsat 5/7/8 EVI time series over cloud-prone, fragmented and mosaic agricultural landscapes is still a great challenge due to low density of high-quality Landsat EVI observations. In those agricultural landscapes, the percentage of high-quality Landsat EVI observations was very low, less than 30% in Haean basin in 2009. VI temporal interpolation functions that fit solely to the Landsat VI data has yielded large RMSE in EVI reconstruction over 0.1 (Pouliot and Latifovic, 2018; Liu et al., 2017). To improve reliability of EVI reconstruction accuracy by using the LTDG_DL, we suggested development of the endmemeber library prior to application of the LTDG_DL algorithm at a given agro-climatic zone. The endmemeber library included time series of Landsat EVI observations in crop pixels. The landscape-level DOYini values can be determined by the onset of the consistent increase phase of the EVI seasonal observations in fifty randomly selected pixels. The endmember EVI library plus a prior knowledge method help to visually interpret the landscape-level DOYini by the onset of the consistent increase phase of the EVI seasonal curves collected from a certain number of crop pixels across a research zone. The parameter setting of the LTDG_DL algorithm was described in detail in Materials and Methods as well as Discussion.

Reference

Pouliot, D., Latifovic, R., 2018. Reconstruction of Landsat time series in the presence of irregular and sparse observations: Development and assessment in north-eastern Alberta, Canada. Remote Sensing of Environment 204, 979–996.

Liu, R., Set al., 2017. Global evaluation of gap-filling approaches for seasonal NDVI with considering vegetation growth trajectory, protection of key point, noise resistance and curve stability. Remote Sensing of Environment 189, 164–179.

4) The methodology section and the validation was little difficult to understand for me and it required multiple reading. Please rewrite the text to make it simpler to the reader.

Reply: We invited a native English speaker to polish the manuscript. We highlighted all changes with marked labels.

5) The figures are unreadable. Similarly, some notations (e.g. line 301 in page 10) embedded as figures within text are not clear. Make them readable and use high resolution figures.

Reply: Thanks for your suggestions. All figures involved in the revised manuscript have been updated using high resolution versions. Furthermore, we added geophysical map of Haean basin and Shandong province in figure 4 where Landsat EVI reconstruction experiments were carried out. Please kindly check them.

Comments on the Quality of English Language

As mentioned before, please simplify the methodology section and the results section for easy understanding.

Reply: We invited a native English speaker to polish the manuscript. We highlighted all changes with marked labels.

Round 2

Reviewer 1 Report

The revised manuscript can be published in Remote Sensing